# A Communication-efficient Algorithm with Linear Convergence for Federated Minimax Learning

**Zhenyu Sun**
Department of Electrical and Computer Engineering
Northwestern University
Evanston, IL 60208
`zhenyusun2026@u.northwestern.edu`

**Ermin Wei**
Department of Electrical and Computer Engineering
Northwestern University
Evanston, IL 60208
`ermin.wei@northwestern.edu`

## Abstract

In this paper, we study a large-scale multi-agent minimax optimization problem, which models many interesting applications in statistical learning and game theory, including Generative Adversarial Networks (GANs). The overall objective is a sum of agents' private local objective functions. We focus on the federated setting, where agents can perform local computation and communicate with a central server. Most existing federated minimax algorithms either require communication per iteration or lack performance guarantees with the exception of Local Stochastic Gradient Descent Ascent (SGDA), a multiple-local-update descent ascent algorithm which guarantees convergence under a diminishing stepsize. By analyzing Local SGDA under the ideal condition of no gradient noise, we show that generally it cannot guarantee exact convergence with constant stepsizes and thus suffers from slow rates of convergence. To tackle this issue, we propose FedGDA-GT, an improved Federated (Fed) Gradient Descent Ascent (GDA) method based on Gradient Tracking (GT). When local objectives are Lipschitz smooth and strongly-convex-strongly-concave, we prove that FedGDA-GT converges linearly with a constant stepsize to global $\epsilon$-approximation solution with $\mathcal{O}(\log(1/\epsilon))$ rounds of communication, which matches the time complexity of centralized GDA method. Then, we analyze the general distributed minimax problem from a statistical aspect, where the overall objective approximates a true population minimax risk by empirical samples. We provide generalization bounds for learning with this objective through Rademacher complexity analysis. Finally, we numerically show that FedGDA-GT outperforms Local SGDA.

## 1 Introduction

In recent years, minimax learning theory has achieved significant success in attaching relevance to many modern machine learning and statistical learning frameworks, including Generative Adversarial Networks (GANs) [1–3], reinforcement learning [4], adversarial training [5, 6], robust estimation and optimization [7–11], and domain adaptation [12, 13]. Generally, a minimax learning problem is modeled as a game between two players with opposite goal, i.e., one minimizes the objective while the other maximizes it.

36th Conference on Neural Information Processing Systems (NeurIPS 2022).

Most current studies in the field of machine learning are targeted at understanding the minimax problem from the view of the speed of convergence and the accuracy of fixed points. In the centralized setting, gradient descent ascent (GDA), which is an extension of gradient descent (GD), stands out for its simple implementation. Specifically, at each iteration, the "min" player conducts gradient descent over its decision variable while the "max" player performs gradient ascent in contrast. Due to the huge volumn of data, stochastic gradient descent ascent (SGDA) is preferred in machine learning settings. Theoretical guarantees are well-established for GDA and SGDA in [14, 15]. However, in practical scenarios, concerns on computation efficiency and data privacy trigger the development of federated learning over a server-client topology and distributed learning over a general graph. These often require communication with the server or neighbors at each iteration [13, 16, 17] and inapplicable to scenarios where communication is expensive.

In this work, we focus on solving a minimax problem in Federated Learning (FL) setting with one server and multiple client/agents. In FL, the server hands over computation burden to agents, which perform training algorithms on their local data. The local trained models are then reported to the server for aggregation. This process is repeated with periodic communication. Much existing literature in FL, however, focuses on minimization optimization [18–22]. The limited literature on federated minimax problems either lacks theoretical guarantees or requires frequent communication [13, 23, 24], with the exception of Local Stochastic Gradient Descent Ascent (SGDA) [25]. In Local SGDA, each agent (or client) performs multiple steps of stochastic gradient descent ascent before communication to the server, which then aggregates local models by averaging. Under careful selection of diminishing learning rates, [25, 26] show that Local SGDA converges to the global optimal solution sub-linearly. However, as we show here, when we try to improve the speed of convergence and reduce communication overhead by introducing a constant stepsize to Local SGDA, it fails to converge to the exact optimal solution, even when full gradients are used. Thus Local SGDA can either be fast(with little communication)-but-inaccurate or accurate-but-slow(with much communication). To address this tradeoff between model accuracy and communication efficiency, we develop FedGDA-GT, Federated Gradient Descent Ascent based on Gradient Tracking (FedGDA-GT), and show that it can achieve fast linear convergence while preserving accuracy.

In addition to solving the minimax problem, we also study the generalization performance of distributed minimax problems in statistical learning, which measures the influence of sampling on the trained model. In most existing works, generalization analysis is established in the context of empirical risk minimization (ERM) [27, 29]. It is well-known that for generic loss functions, the learning error for centralized ERM is on the order of $\mathcal{O}(1/\sqrt{N})$ with $N$ denoting the total number of training samples. Recently, several works derive generalization bounds on centralized minimax learning problems with the same order [30–32]. For generalization analysis in distributed minimax learning, learning bounds are only provided for specific scenarios, e.g., agnostic federated learning [13] and multiple-source domain adaptation [12]. In this paper, we provide generalization bounds for distributed empirical minimax learning with the same order as results of centralized cases, generalizing the results in [13].

## 1.1 Related work

**Centralized minimax learning**  Historically, minimax problems have gained attraction of researchers since several decades ago. An early instantiation is bilinear minimax problem, which becomes a milestone in game theory together with von Neumann's theorem [33]. A simple algorithm is then proposed to solve this bilinear problem efficiently [34]. [35] generalizes von Neumann's theorem to convex-concave games, which triggers an explosion in algorithmic research [36–38]. GDA, as one widely used algorithm, is notable for its simple implementation. It is well-known that GDA can achieve an $\epsilon$-approximation solution with $\mathcal{O}(\log(1/\epsilon))$ iterations for strongly-convex-strongly-concave games and with $\mathcal{O}(\epsilon^{-2})$ iterations for convex-concave games under diminishing stepsizes [14]. Very recently, nonconvex-nonconcave minimax problems appear to be a main focus in optimization and machine learning, due to the emergence of GANs. Several related works are listed therein [39–43].

**Distributed and federated minimax learning**  A few recent studies are devoted to distributed minimax problems due to the increasing volume of data and concerns on privacy and security. Algorithm design and convergence behaviors are extensively studied for minimax problems in the context of distributed optimization, where communication is required at each iteration [44–47].

In the federated setting, [48] proposes a multiple-local-update algorithm to deal with distribution shift issue. [49] studies federated adversarial training under nonconvex-PL objectives. FedGAN is proposed in [50] to train GANs in a communication-efficient way. However, these works are targeted at some specific scenarios. Very recently, aiming to solve the general federated minimax problems, [25] proposes Local SGDA by allowing each agent performing multiple steps of GDA before communication. The authors also prove sub-linear convergence for Local SGDA under diminishing stepsizes. Their convergence guarantees is then improved by [26] to match the results of centralized SGDA [15]. However, we note that all these algorithms require diminishing learning rates to obtain exact solutions, which suffer from relatively slow convergence speed, but our algorithm allows constant stepsizes and hence linear convergence can be achieved.

**Generalization of minimax learning** Recently, generalization properties of minimax learning problems have been widely studied in different scenarios, including GANs and adversarial training. For GANs, [51] analyzes the generalization performance of GANs when discriminators have restricted approximability. [52] evaluates generalization bounds under different metrics. In contrast, [53] suggests a dilemma about GANs' generalization properties. In the context of adversarial training, generalization performances are studied through Rademacher complexity analysis [54, 57], function transformation [55], margin-based [56] approaches. [58] studies the generalization bounds of convex-concave objective functions with Lipschitz continuity. However, all these works are under centralized setting. Recently, [13] provides generalization analysis under agnostic federated learning, where the objective is optimized for any target distribution formed by a mixture of agents' distributions. Our work extends their generalization analysis to general distributed minimax learning problems.

## 1.2 Our Contributions.

We summarize our main contributions as follows: (1) In federated setting, characterizing the behavior of fixed points of Local SGDA, which reveals the impact of objective heterogeneity and multiple local updates on the model accuracy [see Section 3.1]; (2) Resolving the tradeoff between model accuracy and communication efficiency by developing a linear-rate federated minimax algorithm that guarantees exact convergence [see Section 3.2]; (3) Analyzing the generalization properties of empirical minimax learning in distributed settings through Rademacher complexity analysis [see Section 4]; (4) Providing numerical results which suggest communication efficiency of our algorithm compared to Local SGDA and centralized GDA [see Section 5].

**Notations.** In this paper, we let $\| \cdot \|$ denote $l_2$-norm and $| \cdot |$ denote the cardinality of a set or a collection, or absolute value of a scalar. Vectors are column vectors by default and $z = (x, y)$ forms the concatenated vector with $z = [x^T, y^T]^T$. Vectors and scalars for agent $i$ are denoted using subscript $i$, e.g., $f_i(x, y)$. Superscripts, e.g., $t$, denote the indices of iterations. We let the gradient of $f(x, y)$ by $\nabla f(x, y) = (\nabla_x f(x, y), \nabla_y f(x, y))$, where $\nabla_x f(x, y)$ and $\nabla_y f(x, y)$ denote the gradients with respect to $x$ and $y$, respectively.

## 2 Problem Setup

In this paper, we consider the general constrained minimax distributed optimization problem collectively solved by $m$ agents:

$$\min_{x \in \mathcal{X}} \max_{y \in \mathcal{Y}} \left\{ f(x, y) := \frac{1}{m} \sum_{i=1}^{m} f_i(x, y) \right\}, \tag{1}$$

where $\mathcal{X}, \mathcal{Y}$ are some compact feasible sets contained in $\mathbb{R}^p$ and $\mathbb{R}^q$, $x \in \mathcal{X}$ is a $p$-dimension vector, $y \in \mathcal{Y}$ is a $q$-dimension vector and $f_i(\cdot, \cdot)$ is the local objective function of agent $i$, $\forall i = 1 \ldots, m$.

Solving (1) is equivalent to finding a minimax point of $f(x, y)$, defined as follows:

**Definition 1.** *The point $(x^*, y^*)$ is said to be a minimax point of $f(x, y)$ if*

$$f(x^*, y) \leq f(x^*, y^*) \leq \max_{y' \in \mathcal{Y}} f(x, y'), \forall x \in \mathcal{X}, y \in \mathcal{Y}.$$

The first-order necessary condition for minimax points is given by the following lemma.

**Lemma 1.** *[59] Assume $f$ is continuously differentiable. Then, any minimax point $(x^*, y^*)$ in the interior of $\mathcal{X} \times \mathcal{Y}$ satisfies*

$$\nabla_x f(x^*, y^*) = \nabla_y f(x^*, y^*) = 0.$$

# 3 FedGDA-GT: A linear-rate algorithm for federated minimax learning

In this section, we focus on solving (1) in the federated setting, where the $m$ agents are connected to a central server. In general, the agents' communication with the server is more expensive than local computation. We show that the existing methods either require lots of communication (i.e. the convergence rate is sublinear) or could only converge to an inexact solution with linear rates. This suggests a tradeoff between model accuracy and communication efficiency. Motivated by this phenomenon, we aim at developing a communication-efficient minimax algorithm with linear convergence that preserves model accuracy and low communication overhead simultaneously.

We adopt the following standard assumptions on the problem.

**Assumption 1** ($\mu$-strongly-convex-strongly-concave). *For any $i = 1, \ldots, m$, $f_i(x, y)$ is twice-differentiable and is $\mu$-strongly-convex-strongly-concave with some $\mu > 0$ for any $(x, y) \in \mathbb{R}^p \times \mathbb{R}^q$, i.e.,*

$$\text{for any given } y, \qquad f_i(z, y) \geq f_i(x, y) + \langle \nabla_x f_i(x, y), z - x \rangle + \frac{\mu}{2}\|z - x\|^2, \; \forall z, x,$$

$$\text{for any given } x, \qquad f_i(x, z) \leq f(x, y) + \langle \nabla_y f_i(x, y), z - y \rangle - \frac{\mu}{2}\|z - y\|^2, \; \forall z, y.$$

**Assumption 2** ($L$-smoothness). *There exists some $L > 0$ such that for any $i = 1, \ldots, m$, $\|\nabla f_i(x, y) - \nabla f_i(x', y')\| \leq L\|(x, y) - (x', y')\|, \forall (x, y), (x', y') \in \mathbb{R}^p \times \mathbb{R}^q$.*

We note that although each $f_i(x, y)$ may have different $\mu_i$ and $L_i$, we can set $\mu = \min_{i=1,\ldots,m} \mu_i$ and $L = \max_{i=1\ldots,m} L_i$ to ensure Assumptions 1 and 2 hold.

## 3.1 Analysis on Local SGDA

We first study Local Stochastic Gradient Descent Ascent (SGDA) proposed in [25], which is the only known method with convergence guarantees and can utilize multiple local computation steps before communicating with the central server for solving general federated minimax problems. In particular, in each iteration of Local SGDA, each agent updates its local model $(x_i, y_i)$ for $K$ times by using local stochastic gradients before communication, and then sends its local model to the server, which then computes the average of local models and sends it back.

In [25], the authors prove that under Assumptions 1-2, with bounded variance assumption on all local gradients and vanishing learning rate, after $T$ rounds of communication, the convergence result of Local SGDA is

$$\mathbb{E}\left[\|x^{(T)} - x^*\|^2 + \|y^{(T)} - y^*\|^2\right] \leq \mathcal{O}\left(T^{-1}\right) + \mathcal{O}\left(T^{-3}\right). \tag{2}$$

This shows the slow sublinear convergence rate of this method, which translates to large amount of communication. We next consider an ideal deterministic version of Local SGDA (Algorithm 1), where local stochastic gradients are replaced with full gradients.

---

**Algorithm 1** Local SGDA

**Input:** $(x^0, y^0)$ as initialization of the server
1: **for** $t = 0, 1, \ldots, T$ **do**
2: $\quad x_{i,0}^{t+1} = x^t, \quad y_{i,0}^{t+1} = y^t, \; \forall i = 1, \ldots, m$
3: $\quad$ **for** $k = 0, 1, \ldots, K - 1$ **do** (in parallel for all agents)
4: $\quad\quad x_{i,k+1}^{t+1} = x_{i,k}^{t+1} - \eta_x \nabla_x f_i(x_{i,k}^{t+1}, y_{i,k}^{t+1})$
5: $\quad\quad y_{i,k+1}^{t+1} = y_{i,k}^{t+1} + \eta_y \nabla_y f_i(x_{i,k}^{t+1}, y_{i,k}^{t+1})$
6: $\quad$ **end for**
7: $\quad x^{t+1} = \frac{1}{m}\sum_{i=1}^m x_{i,K}^{t+1}, \quad y^{t+1} = \frac{1}{m}\sum_{i=1}^m y_{i,K}^{t+1}$
8: **end for**
**Output:** $(x^T, y^T)$ given by the server

---

In order to rewrite Local SGDA in a concise form, we define the operator for gradient descent evaluated at $(x, y)$ for agent $i$ as

$$\mathcal{D}_i^1(x, y) = x - \eta_x \nabla_x f_i(x, y), \; \forall i = 1, \ldots, m.$$

Given some $k \geq 1$, let $\mathcal{D}_i^k$ define the composition of $k$ $\mathcal{D}_i^1$ operators. Moreover, $\mathcal{D}_i^0(x, y) = x$ is the identity operator on the first argument. Let $\mathcal{A}_i^1(x, y) = y + \eta_y \nabla_y f_i(x, y)$ be the gradient ascent operator with $\mathcal{A}_i^k(x, y)$ and $\mathcal{A}_i^0(x, y) = y$ defined similarly. Then, given some initial point $(x^0, y^0)$, Algorithm 1 is rewritten as the recursion of the following for $t = 0, 1, \ldots, T$:

$$\tilde{x}_i^{t+1} = \mathcal{D}_i^K(x^t, y^t), \quad \tilde{y}_i^{t+1} = \mathcal{A}_i^K(x^t, y^t), \quad \forall i = 1, \ldots, m$$

$$x^{t+1} = \frac{1}{m} \sum_{i=1}^m \tilde{x}_i^{t+1}, \quad y^{t+1} = \frac{1}{m} \sum_{i=1}^m \tilde{y}_i^{t+1}. \tag{3}$$

The following result shows that Local SGDA generally cannot guarantee the convergence to the optimal solution of (1) with fixed stepsizes even with full gradients.

**Proposition 1.** *Suppose $f_i$ is differentiable, $\forall i = 1, \ldots, m$. For any $K \geq 1$, let $\{(x^t, y^t)\}$ be the sequence generated by (3) (or equivalently by Algorithm 1). If $\{(x^t, y^t)\}$ converges to a fixed point $(x^*, y^*)$, then*

$$\frac{1}{m} \sum_{i=1}^m \sum_{k=0}^{K-1} \nabla f_i(\mathcal{D}_i^k(x^*, y^*), \mathcal{A}_i^k(x^*, y^*)) = 0.$$

Note that when $K = 1$, Local SGDA reduces to GDA and the fixed point $(x^*, y^*)$ satisfies

$$\frac{1}{m} \sum_{i=1}^m \nabla_x f_i(x^*, y^*) = \frac{1}{m} \sum_{i=1}^m \nabla_y f_i(x^*, y^*) = 0$$

which meets the optimality condition (by Lemma 1) of the minimax point of $f(x, y)$. For any other $K$, the fixed point characterization is different from the minimax point. An illustrative example is provided in Appendix C.

We mention that while having multiple local gradient steps in the inner loop allows the agents to reduce the frequency of communication, it also pulls the agents towards their local minimax points as opposed to the global one. Motivated by this observation, we next propose a modification involving a gradient correction term, also known as gradient tracking (GT), to make sure the agents are moving towards the global minimax point.

## 3.2 FedGDA-GT and convergence guarantees

As indicated in Section 3.1, Local SGDA cannot guarantee exact convergence under fixed stepsizes, which implies a tradeoff between communication efficiency and model accuracy. Aiming at resolving this issue, in this section, we introduce our algorithm FedGDA-GT, formally described in Algorithm 2, that can reach the optimal solution of (1) with fixed stepsizes, full gradients and the following assumption.

**Assumption 3.** *The feasible sets $\mathcal{X}$ and $\mathcal{Y}$ are compact and convex. Moreover, there is at least one minimax point of $f(x, y)$ lying in $\mathcal{X} \times \mathcal{Y}$.*

$\text{Proj}_{\mathcal{X}}(\cdot)$ and $\text{Proj}_{\mathcal{Y}}(\cdot)$ denote the projection operators, i.e.,

$$\text{Proj}_{\mathcal{X}}(x) = \arg\min_{z \in \mathcal{X}} \|z - x\| \text{ and } \text{Proj}_{\mathcal{Y}}(y) = \arg\min_{z \in \mathcal{Y}} \|z - y\|,$$

which are well defined by the previous assumption and are used to guarantee the output of Algorithm 2 is feasible. The main differences between FedGDA-GT and Local-SGDA are the correction terms of $-\nabla_x f_i(x^t, y^t) + \nabla_x f(x^t, y^t)$ and $-\nabla_y f_i(x^t, y^t) + \nabla_y f(x^t, y^t)$, which track the differences between local and global gradients. By including the gradient correction terms, when $(x_{i,k}^{t+1}, y_{i,k}^{t+1})$ is not too far from $(x^t, y^t)$, we can expect $\nabla_x f_i(x_{i,k}^{t+1}, y_{i,k}^{t+1}) \approx \nabla_x f_i(x^t, y^t)$ and similarly for $y$ gradients. Hence the updates reduce to simply taking global gradient descent and ascent steps, which coincide with centralized GDA updates and thus would have the correct fixed points. We next show the convergence guarantee of FedGDA-GT.

We first establish the uniqueness of minimax point of $f(x, y)$ by the following lemma.

---

**Algorithm 2** FedGDA-GT

---

**Input:** $(x^0, y^0)$ as initialization of the global model
1: **for** $t = 0, 1, \ldots, T$ **do**
2:     Server broadcasts $(x^t, y^t)$
3:     Agents compute $(\nabla_x f_i(x^t, y^t), \nabla_y f_i(x^t, y^t))$ and send it to the server
4:     Server computes $(\nabla_x f(x^t, y^t), \nabla_y f(x^t, y^t))$ and broadcasts it
5:     Each agent $i$ for $i = 1, \ldots, m$ sets $x_{i,0}^{t+1} = x^t$, $y_{i,0}^{t+1} = y^t$
6:     **for** $k = 0, 1, \ldots, K - 1$ **do** (in parallel for all agents)
7:         $x_{i,k+1}^{t+1} = x_{i,k}^{t+1} - \eta(\nabla_x f_i(x_{i,k}^{t+1}, y_{i,k}^{t+1}) - \nabla_x f_i(x^t, y^t) + \nabla_x f(x^t, y^t))$
8:         $y_{i,k+1}^{t+1} = y_{i,k}^{t+1} + \eta(\nabla_y f_i(x_{i,k}^{t+1}, y_{i,k}^{t+1}) - \nabla_y f_i(x^t, y^t) + \nabla_y f(x^t, y^t))$
9:     **end for**
10:    All agents send $(x_{i,K}^{t+1}, y_{i,K}^{t+1})$ to the server to compute $(x^{t+1}, y^{t+1})$ by
11:    $x^{t+1} = \text{Proj}_{\mathcal{X}} \left( \frac{1}{m} \sum_{i=1}^m x_{i,K}^{t+1} \right), \quad y^{t+1} = \text{Proj}_{\mathcal{Y}} \left( \frac{1}{m} \sum_{i=1}^m y_{i,K}^{t+1} \right)$
12: **end for**
**Output:** $(x^T, y^T)$ given by the server

---

**Lemma 2.** *Under Assumptions 1 and 3, $(x^*, y^*)$ is the unique minimax point of $f(x, y)$ if and only if*

$$\nabla_x f(x^*, y^*) = \nabla_y f(x^*, y^*) = 0.$$

Next, we state the main convergence result of our algorithm.

**Theorem 1.** *Suppose Assumptions 1, 2, 3 are satisfied. Let $\{(x^t, y^t)\}_{t=0}^\infty$ be a sequence generated by Algorithm 2. Then there exists a scalar $\eta_0 > 0$, such that for any stepsize $\eta \in (0, \eta_0)$*

$$\|x^t - x^*\|^2 + \|y^t - y^*\|^2 \leq \rho(\eta)^t \left( \|x^0 - x^*\|^2 + \|y^0 - y^*\|^2 \right), \forall t = 0, 1, \ldots,$$

*where $(x^*, y^*)$ is the unique minimax point of $f(x, y)$ and $\rho(\eta)$ is some scalar in $(0, 1)$.*

Theorem 1 guarantees linear convergence of FedGDA-GT (Algorithm 2) to the correct optimal solution of (1) under suitable choices of stepsize $\eta$. We note that linear convergence here is with respect to the outer-loop (indexed by $t$), as the inner-loop (indexed by $k$) can be implemented cheaply without any communication. The fast convergence speed and exact convergence of this result eliminate the tradeoff between model accuracy and communication efficiency. Furthermore, no restriction on heterogeneity level of local objectives is placed. For the homogeneous setting, we show in Appendix D.4 that the convergence rate of FedGDA-GT can be improved at least $K$ times, compared to heterogeneous setting.

## 4 Generalization bounds on minimax learning problems

In this section, we consider minimax statistical learning, an important application of the minimax framework. We view the problem (1) in Section 3 as the empirical version of a population minimax optimization problem. We evaluate the generalization performance of the empirical problem using Rademacher complexity.

To be more specific, in a minimax statistical learning task, each agent is allocated with some local dataset $\mathcal{S}_i = \{\xi_{i,j}\}_{j=1}^{n_i}$, where $\xi_{i,j}$ denotes the $j$th sample of agent $i$ and $n_i$ denotes the number of local samples. $\mathcal{S} = \bigcup_{i=1}^m \mathcal{S}_i$ is the dataset with all samples. Moreover, we assume $n_i = n, \forall i = 1, \ldots, m$ and $N = mn$ is the total number of samples. Then, each local objective function $f_i(x, y)$ is defined by using local data, i.e.,

$$f_i(x, y) = \frac{1}{n} \sum_{j=1}^n l(x, y; \xi_{i,j}), \tag{4}$$

where $l(x, y; \xi_{i,j})$ is the loss function measured at point $\xi_{i,j}$ and (4) is called local empirical minimax risk. Suppose that for agent $i$ each data sample $\xi_{i,j}$ is independently drawn from some underlying distribution $P_i$, denoted by $\xi \sim P_i$. We can further define the local population minimax risk as follows:

$$R_i(x, y) = \mathbb{E}_{\xi \sim P_i} [l(x, y; \xi)] \tag{5}$$

and similarly, the global population minimax risk is defined by

$$R(x, y) = \mathbb{E}_{\xi \sim P}\left[l(x, y; \xi)\right], \tag{6}$$

where $P$ is the underlying distribution of the whole dataset $\mathcal{S}$. Then, the minimax problem based on the population minimax risk is given by

$$\min_{x \in \mathcal{X}} \max_{y \in \mathcal{Y}} R(x, y). \tag{7}$$

Sample applications of our formulation (4)-(7) can be found in Appendix A.

While we are interested in calculating the population minimax risk, in practice we can only solve (1), which is a sampled version of (7), since the true distribution $P$ is unknown. In this sense, how could we expect the optimal solution of (1) performs successfully on (6)? To answer this question, we provide the generalization bound which measures the performance of the model trained on the empirical minimax risk $f(x, y)$. Our generalization bound is based on the notion of Rademacher complexity [29] defined by

$$\mathscr{R}(\mathcal{X}, y) = \mathbb{E}_{\xi \sim P}\mathbb{E}_{\sigma}\left[\sup_{x \in \mathcal{X}} \frac{1}{mn} \sum_{i=1}^{m} \sum_{j=1}^{n} \sigma_{i,j} l(x, y; \xi_{i,j})\right], \tag{8}$$

where $\sigma = \{\sigma_{i,j}\}, \forall i = 1, \ldots, m, \forall j = 1, \ldots, n$, is a collection of Rademacher variables taking values from $\{-1, 1\}$ uniformly. Basically, the Rademacher complexity (8) captures the capability of $\mathcal{X}$ to fit random sign noise, i.e., $\sigma$. Note that $\sigma_{i,j} l(x, y; \xi_{i,j})$ measures how well the loss $l$ correlates with noise $\sigma$ on sample space $\mathcal{S}$. By taking the supremum, it means what the best extent the feasible set $\mathcal{X}$ on average can correlate with random noise $\sigma$. Thus, if $\mathcal{X}$ is richer, then the Rademacher complexity is bigger. Moreover, the minimax Rademacher complexity is defined on $\mathcal{Y}$ by

$$\mathscr{R}(\mathcal{X}, \mathcal{Y}) = \max_{y \in \mathcal{Y}} \mathscr{R}(\mathcal{X}, y). \tag{9}$$

Given $\epsilon > 0$, we further define the $\epsilon$-minimum cover of $\mathcal{Y}$ in $l_2$ distance by

$$\mathcal{Y}_\epsilon = \arg \min_{C(\mathcal{Y}, \epsilon)} \left|C(\mathcal{Y}, \epsilon)\right|,$$

where $C(\mathcal{Y}, \epsilon) = \{B_\epsilon(y) : y \in \mathcal{Y}\}$ is a collection of open balls $B_\epsilon(y)$ centered at $y$ with radius $\epsilon$ such that for any $y \in \mathcal{Y}$ there exists some $B_\epsilon(y') \in C(\mathcal{Y}, \epsilon)$ with $y \in B_\epsilon(y')$. Note that $\left|\mathcal{Y}_\epsilon\right| < \infty$, since $\mathcal{Y}$ is compact and thus every open cover of $\mathcal{Y}$ has a finite subcover.

Then, we have the following generalization bound for the distributed minimax learning problem:

**Theorem 2.** *Suppose* $|l(x, y; \xi) - l(x, y'; \xi)| \le L_y\|y - y'\|$ *and* $|l(x, y; \xi)| \le M_i(y)$, $\forall x \in \mathcal{X}$, $\forall y, y' \in \mathcal{Y}_\epsilon$ *and* $\forall \xi \sim P_i, i = 1 \ldots, m$ *with some positive scalar* $L_y$ *and real-valued function* $M_i(y) > 0$. *Then, given any* $\epsilon > 0$ *and* $\delta > 0$, *with probability at least* $1 - \delta$ *for any* $(x, y) \in \mathcal{X} \times \mathcal{Y}$,

$$R(x, y) \le f(x, y) + 2\mathscr{R}(\mathcal{X}, y) + \sqrt{\sum_{i=1}^{m} \frac{M_i^2(y)}{2m^2 n} \log \frac{|\mathcal{Y}_\epsilon|}{\delta}} + 2L_y\epsilon. \tag{10}$$

Theorem 2 generally states that given any $x, y$ feasible, it is highly possible that the distance between the global population minimax risk $R(x, y)$ and the global empirical minimax risk $f(x, y)$ can be bounded by Rademacher complexity and a term related to the number of agents and local sample size. Moreover, we allow the upper bound of $l(\cdot, y)$ can depend on different choices of $y$ and agents since different local distributions always have different effects on the value of $l(\cdot, y)$.

Based on Theorem 2, we further derive the high-probability bound on $\max_{y \in \mathcal{Y}} R(x, y)$.

**Corollary 1.** *Under the same conditions of Theorem 2, with probability at least* $1 - \delta$ *for any* $x \in \mathcal{X}$, *the following inequality holds for any* $\epsilon > 0$, $\delta > 0$:

$$Q(x) \le g(x) + 2\mathscr{R}(\mathcal{X}, \mathcal{Y}) + \sqrt{\max_{y \in \mathcal{Y}} \left\{\sum_{i=1}^{m} \frac{M_i^2(y)}{2m^2 n}\right\} \log \frac{|\mathcal{Y}_\epsilon|}{\delta}} + 2L_y\epsilon \tag{11}$$

*where* $Q(x) = \max_{y \in \mathcal{Y}} R(x, y)$, $g(x) = \max_{y \in \mathcal{Y}} f(x, y)$ *are the worst-case population and empirical risks, respectively.*

We can further bound $\mathscr{R}(\mathcal{X}, \mathcal{Y})$ when the loss function $l(x, y; \xi)$ takes finite number of values for any $(x, y) \in \mathcal{X} \times \mathcal{Y}$ and any data samples.

**Lemma 3.** *Suppose for any $y \in \mathcal{Y}$ and $i \in \{1, \ldots, m\}$, $|l(\cdot, y; \cdot)|$ is bounded by $M_i(y)$ and takes finite number of values. Further, assume that the VC-dimension of $\mathcal{X}$ is $d$. Then, the following inequality holds:*

$$\mathscr{R}(\mathcal{X}, \mathcal{Y}) \leq \sqrt{2d \max_{y \in \mathcal{Y}} \left\{ \sum_{i=1}^{m} \frac{M_i^2(y)}{m^2 n} \right\} \left( 1 + \log \frac{mn}{d} \right)}. \tag{12}$$

Corollary 1 measures the error between the worst-case population and empirical risks, that is $\max_{y \in \mathcal{Y}} R(x, y) - \max_{y \in \mathcal{Y}} f(x, y)$. A smaller error indicates a better performance of the model, which is trained empirically, generalized on the underlying distribution $P$. Combining Lemma 3, we know that this worst-case error can be bounded by some decreasing function with respect to sample size. Thus, in order to get a better generalization performance, one effective way is to draw more local samples for each agent.

It it worth noting that the generalization bound is related to the term $\sum_{i=1}^{m} M_i^2(y)$ as shown in (10)-(12), which essentially measures the effect of feasible set $\mathcal{Y}$ by means of different local data distributions on each agent. This also captures the heterogeneity of agents.

In fact, the bounds (10)-(12) we proposed are generalized versions of those in [13], which essentially include them as special cases by selecting suitable $M_i(y)$. Specifically, noting that in [13], the global population risk takes the form of $R(x, y) = \sum_{i=1}^{m} y_i R_i(x)$ with $y_i \geq 0$ and $\sum_{i=1}^{m} y_i = 1$, then by choosing $M_i(y) = m y_i M$ with some $M > 0$ we recover the same result. Moreover, compared to [30, 31], where only centralized minimax learning problems are considered, our bounds have the same order of complexity $\mathcal{O}(1/\sqrt{N})$ with $N$ denoting the total sample size by taking the uniform bound on $l(\cdot)$ for all agents.

## 5 Experiments

In this section, we numerically measure the performance of FedGDA-GT compared to Local SGDA with full gradients on a personal laptop by solving (1). We consider first perform experiments on quadratic objective functions with $x$ and $y$ uncoupled. Then, we test our algorithm on the robust linear regression problem. In both cases, FedGDA-GT performs much better than Local SGDA with heterogeneous local objectives.

### 5.1 Uncoupled quadratic objective functions

We first consider the following local objective functions:

$$f_i(x, y) = \frac{1}{2} x^T A_i^T A_i x - \frac{1}{2} y^T A_i^T A_i y + (A_i^T b_i)^T (2x - y), \forall i = 1, \ldots, m, \tag{13}$$

where $x, y \in \mathbb{R}^d$ and $A_i \in \mathbb{R}^{n_i \times d}$ with $n_i$ representing the number of samples of agent $i$. We generate $A_i, b_i$ as follows:

For each agent, every entry of $A_i$, denoted by $[A_i]_{kl}$, is generated by Gaussian distribution $\mathcal{N}(0, (0.5i)^{-2})$. To construct $b_i$, we generate a random reference point $\theta_i \in \mathbb{R}^d$, where $\theta_i \sim \mathcal{N}(\mu_i, I_{d \times d})$. Each element of $\mu_i$ is drawn from $\mathcal{N}(\alpha, 1)$ with $\alpha \sim \mathcal{N}(0, 100)$. Then $b_i = A_i \theta_i + \epsilon_i$ with $\epsilon_i \sim \mathcal{N}(0, 0.25 I_{n_i \times n_i})$. We set the dimension of model as $d = 50$ and number of samples as $n_i = 500$ and train the models with $m = 20$ agents by Algorithm 1 and Algorithm 2, respectively. In order to compare them, the learning rate is $10^{-4}$ for both algorithms and we choose Local SGDA with $K = 1$, which is equivalent to a centralized GDA, as the baseline. Figure 1 shows the trajectories of Algorithms 1 and 2 under objective functions constructed by (13), respectively. Different numbers of local updates are selected (with $K = 20$ and $K = 50$). In this heterogeneous setting, we can see that FedGDA-GT achieves linear convergence, converging to a more accurate solution with significantly fewer rounds of communication, compared with Local SGDA and centralized GDA. Moreover, our numerical results suggest that Local SGDA may converge to a non-optimal point (optimality gap over $10^4$), which conforms with our Theorem 1.

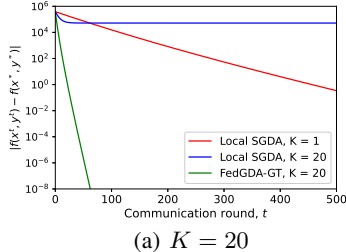
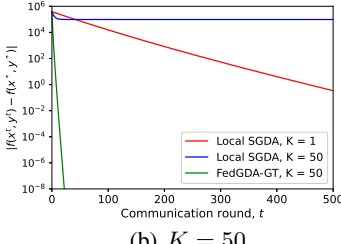

(a) $K = 20$            (b) $K = 50$

Figure 1: Local SGDA and FedGDA-GT with constant stepsizes under different numbers of local updates

## 5.2 Robust linear regression

Next, we consider the problem of robust linear regression, which is widely studied in estimation with gross error [28, 57]. As the same formulation in [25], each agent's loss function is defined by

$$f_i(x,y) = \frac{1}{n_i}\sum_{j=1}^{n_i}(x^T(a_{i,j}+y)-b_{i,j})^2 + \frac{1}{2}\|x\|^2,\ \forall i=1,\ldots,m, \tag{14}$$

where $(a_{i,j}, b_{i,j})$ is the $j$th data sample of agent $i$, $n_i$ is local sample size. Specifically, in (14), $x \in \mathbb{R}^d$ represents the model of linear regression and $y \in \mathbb{R}^d$ represents the gross noise aimed at contaminating each sample. We assume that there is an upper bound on the noise, i.e., $\|y\| \le 1$. By solving $\min_{x \in \mathbb{R}^d} \max_{\|y\| \le 1} \frac{1}{m}\sum_{i=1}^m f_i(x,y)$, we obtain a global robust model of the linear regression problem even under the worst contamination of gross noise. To measure the convergence of algorithms, we use the robust loss, i.e., given a model $\hat{x}$, the corresponding robust loss [25, 26] is defined by $\tilde{f}(\hat{x}) = \max_{\|y\| \le 1} \sum_{i=1}^m f_i(\hat{x},y)$.

We generate local models and data as follows: the local model $x_i^*$ is generated by a multivariate normal distribution. The output for agent $i$ is given by $b_{i,j} = (x_i^*)^T a_{i,j} + \epsilon_j$ with $\epsilon_j \sim \mathcal{N}(0,1)$. Each input point $a_{i,j}$ is with dimension $d$ and drawn from a Gaussian distribution $a_{i,j} \sim \mathcal{N}(\mu_i, K_i)$ where $\mu_i \sim \mathcal{N}(c_i, I_{d\times d})$ and $K_i = i^{-1.3} I_{d\times d}$. Each element of $c_i$ is drawn from $\mathcal{N}(0,\alpha^2)$. By choosing different $\alpha$, we control the heterogeneity of local data and hence $f_i(x,y)$.

In this experiment, we compare Algorithms 1 and 2 under different heterogeneity levels, i.e., $\alpha = 1$, $\alpha = 5$ and $\alpha = 20$. For each case, we choose the same constant $\eta$ for both Local SGDA and FedGDA-GT. As shown in Figure 2, when local agents are more heterogeneous, FedGDA-GT performs better than Local SGDA, which lies not only in faster convergence but also smaller robust loss. Specifically, when $\alpha = 1$, two algorithms almost have the same performance. To explain this phenomenon, let us recall FedGDA-GT again. Smaller $\alpha$ essentially means more similar local objectives. In particular, $\alpha = 0$ corresponds to i.i.d. cases. In this sense, the local updates of FedGDA-GT become the same as that in Local SGDA, which indicates similar performance as shown in Figure 2(a).

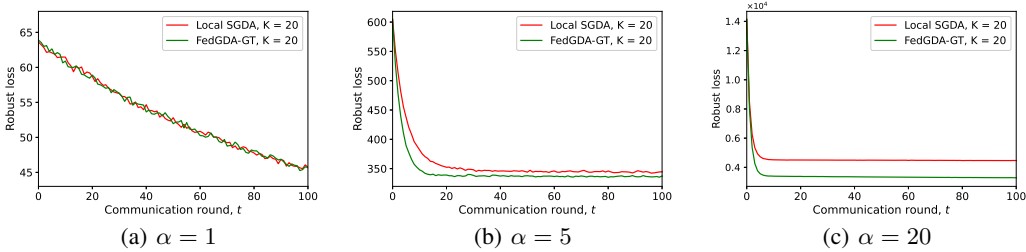

(a) $\alpha = 1$        (b) $\alpha = 5$        (c) $\alpha = 20$

Figure 2: Local SGDA and FedGDA-GT under different heterogeneity levels

## 6 Conclusion

In this paper, we investigate the federated minimax learning problem. We first characterize the fixed-point behavior of a recent algorithm Local SGDA to show that it presents a tradeoff between communication efficiency and model accuracy and cannot achieve linear convergence under constant learning rates. To resolve this issue, we propose FedGDA-GT that guarantees exact linear convergence and reaches $\epsilon$-optimality with $\mathcal{O}(\log(1/\epsilon))$ time, which is the same as centralized GDA method. Then, we study the generalization properties of distributed minimax learning problems. We establish generalization error bounds without strong assumptions on local distributions and loss functions based on Rademacher complexity. The bounds match existing results of centralized minimax learning problems. Finally, we compare FedGDA-GT with two state-of-the-art algorithms, Local SGDA and GDA, through numerical experiments and show that FedGDA-GT outperforms in efficiency and/or accuracy.

## Acknowledgments and Disclosure of Funding

This work was supported by the NSF NRI 2024774.

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
