# A Applications of distributed/federated minimax problems

In this section, we consider specific instantiations of (1) and (7). Two examples are presented: one is federated generative adversarial networks, another is agnostic federated learning. We show that both of them are special cases of the general framework considered in the paper.

## A.1 Federated generative adversarial networks

In [50], the authors consider to train GANs in a federated way, where $m$ agents with corresponding local datasets cooperate to learn a common model which is essentially the model of centralized GAN. Then, for each agent, its local objective function is defined by

$$R_i(x, y) = -\mathbb{E}_{\xi \sim P_i}\left[\log \phi_y(\xi)\right] - \mathbb{E}_{\xi' \sim Q_i(x)}\left[\log(1 - \phi_y(\xi'))\right]$$

where $\phi_y$ is the discriminator and $Q_i(x)$ is the distribution to generate fake data of the generator. And the objective of the centralized GAN is given by

$$R(x, y) = \frac{1}{m}\sum_{i=1}^{m} R_i(x, y)$$

when identical sample sizes are assumed. This is essentially the same as our formulation.

## A.2 Agnostic federated learning

The framework of agnostic federated learning was first proposed and analyzed in [13]. where the centralized model is leaned for any possible target distribution that is formed by a convex combination of all agents' local distributions. In particular, let $P_i$ denote the distribution of agent $i$. Then, the target distribution is formed by $\sum_{i=1}^{m} \lambda_i^* P_i$ for some unknown $\lambda^*$ such that $\lambda^* \in \Lambda$, where $\Lambda$ represents a simplex. Then, agnostic federated learning is aimed at learning a model $\theta^*$ that performs best under the worst case, i.e.,

$$\theta^* = \arg\min_{\theta \in \Theta}\left\{R(\theta, \Lambda) := \max_{\lambda \in \Lambda}\sum_{i=1}^{m} \lambda_i R_i(\theta)\right\}$$

where $R_i(\theta) = \mathbb{E}_{\xi \sim P_i}[l(\theta; \xi)]$ is the local population risk. The empirical version of the problem can be derived similarly. Note that this formulation is essentially included by our problem.

# B Proof of Proposition 1

Given (3), it is straightforward that

$$\begin{aligned}
x_{i,K}^{t+1} &= \mathcal{D}_i^K(x_{i,0}^{t+1}, y_{i,0}^{t+1}), \\
y_{i,K}^{t+1} &= \mathcal{A}_i^K(x_{i,0}^{t+1}, y_{i,0}^{t+1}).
\end{aligned}$$

Noting $(x^{t+1}, y^{t+1}) = \frac{1}{m}\sum_{i=1}^{m}(x_{i,K}^{t+1}, y_{i,K}^{t+1})$ and $(x^t, y^t) = (x_{i,0}^{t+1}, y_{i,0}^{t+1})$,

$$\begin{aligned}
x^{t+1} &= \frac{1}{m}\sum_{i=1}^{m}\mathcal{D}_i^K(x^t, y^t), \\
y^{t+1} &= \frac{1}{m}\sum_{i=1}^{m}\mathcal{A}_i^K(x^t, y^t).
\end{aligned}$$

Further using $\lim_{t\to\infty}(x^t, y^t) = (x^*, y^*)$ gives

$$\begin{aligned}
\frac{1}{m}\sum_{i=1}^{m}\sum_{k=0}^{K-1}\nabla_x f_i(\mathcal{D}_i^k(x^*, y^*), \mathcal{A}_i^k(x^*, y^*)) &= 0, \\
\frac{1}{m}\sum_{i=1}^{m}\sum_{k=0}^{K-1}\nabla_y f_i(\mathcal{D}_i^k(x^*, y^*), \mathcal{A}_i^k(x^*, y^*)) &= 0,
\end{aligned}$$

which completes the proof.

## C  An illustrative example for Local SGDA with constant stepsizes

We illustrate the inexact convergence of local SGDA with constant stepzie through a simple instance of (1) where only two agents cooperate to find a minimax point of $f(x, y)$ by Local SGDA using full gradient information. We further assume that each $f_i(x, y)$ is strongly-convex-strongly-concave and Lipschitz smooth such that the minimax point is unique and linear convergence is possible to reach. Specifically, we construct local objectives as follows:

$$
\begin{aligned}
f_1(x, y) &= x^2 - y^2 - (x - y) \\
f_2(x, y) &= 4x^2 - 4y^2 - 32(x - y)
\end{aligned}
$$

where the minimax point is $x^* = y^* = \left(\sum_{i=1}^{2} 2i^2\right)^{-1} \sum_{i=1}^{2}(31i - 30)$. By Proposition 1, a straightforward calculation gives

$$
\begin{aligned}
x^*_{\text{Local-SGDA}} &= \left(\sum_{i=1}^{2} \sum_{k=0}^{K-1} 2i^2(1 - 2\eta_x i^2)^k\right)^{-1} \sum_{i=1}^{2} \sum_{k=0}^{K-1}(31i - 30)(1 - 2\eta_x i^2)^k, \\
y^*_{\text{Local-SGDA}} &= \left(\sum_{i=1}^{2} \sum_{k=0}^{K-1} 2i^2(1 - 2\eta_y i^2)^k\right)^{-1} \sum_{i=1}^{2} \sum_{k=0}^{K-1}(31i - 30)(1 - 2\eta_y i^2)^k.
\end{aligned}
$$

In general $x^*_{\text{FedGDA}} \neq x^*$, $y^*_{\text{FedGDA}} \neq y^*$ when $K \geq 2$. Therefore, we see that Local SGDA has incorrect fixed points when constant stepsizes are used even under deterministic scenarios.

In the sequel, we empirically show the effect of different numbers of local updates on the fixed point. We consider cases with $K = 1$, $K = 10$, $K = 20$, $K = 50$. The stepsizes $\eta_x$ and $\eta_y$ are set by $\eta_x = \eta_y = 0.1$ when $K = 1$ and by $\eta_x = \eta_y = 0.001$ for the remaining cases. The initial points $(x^0, y^0)$ for four cases are identical for the convenience of comparison. From Figure 3, when $K = 1$ Local SGDA reduces to centralized GDA and converges to the minimax point $(x^*, y^*)$ linearly by strong-convexity-strong-concavity and Lipschitz smoothness assumptions. However, for $K = 10, 20, 50$, given identical stepsizes, larger the number of local updates is, fewer communication rounds are needed until convergence, but farther the limit points are from the optimal one. Another point that is worthy to note is that convergence error between the minimax point and the fixed point of Local SGDA can be too large to be neglected (even over $10^3$ in Section 5), although the errors shown in Figure 3 are relatively small.

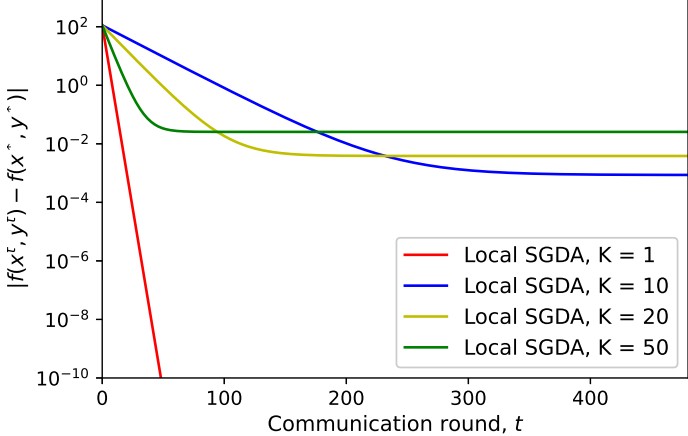

Figure 3: Local SGDA with constant stepsizes under different numbers of local updates

# D   Convergence analysis of FedGDA-GT

## D.1   Proof of Lemma 2

First, we introduce the definition of saddle point of $f(x, y)$:

**Definition 2.** *The point $(x^*, y^*)$ is said to be a saddle point of $f(x, y)$ if*

$$f(x^*, y) \leq f(x^*, y^*) \leq f(x, y^*), \forall x \in \mathcal{X}, y \in \mathcal{Y}.$$

Obviously, by Definitions 1 and 2, we know that any saddle point of $f(x, y)$ is also a minimax point of $f(x, y)$. Then, any saddle point in the interior of $\mathcal{X} \times \mathcal{Y}$ must satisfy Lemma 1. Moreover, when $f(x, y)$ is strongly-convex-strongly-concave, we show that any minimax point is also a saddle point, stated as follows:

**Lemma 4.** *Suppose $f(x, y)$ satisfies Assumptions 1 and 3. If $(x^*, y^*)$ is a minimax point of $f(x, y)$, then it is also a saddle point of $f(x, y)$. Moreover,*

$$\nabla_x f(x^*, y^*) = \nabla_y f(x^*, y^*) = 0 \iff (x^*, y^*) \text{ is a saddle point of } f(x, y).$$

*Proof.* By Assumption 1, we have

$$\nabla_{xx}^2 f(x, y) \succeq \mu I, \quad \nabla_{yy}^2 f(x, y) \preceq -\mu I, \quad \forall x, y.$$

By Lemma 1 and Proposition 5 in [59], we obtain that under Assumptions 1 and 3,

$$\nabla_x f(x^*, y^*) = \nabla_y f(x^*, y^*) = 0 \iff (x^*, y^*) \text{ is a saddle point of } f(x, y).$$

Noting that $(x^*, y^*)$ is a minimax point implies $\nabla_x f(x^*, y^*) = \nabla_y f(x^*, y^*) = 0$ by Lemma 1, this completes the proof. $\square$

Next, we provide the uniqueness statement of saddle point $(x^*, y^*)$.

**Lemma 5.** *Under Assumption 1, the saddle point $(x^*, y^*)$ of $f(x, y)$ is unique in $\mathcal{X} \times \mathcal{Y}$.*

*Proof.* By Assumption 1, it yields given any $y \in \mathbb{R}^q$ and $\alpha \in (0, 1)$,

$$f(\alpha x + (1 - \alpha)z, y) \leq \alpha f(x, y) + (1 - \alpha)f(z, y) - \frac{\mu}{2}\alpha(1 - \alpha)\|z - x\|^2, \quad \forall x, z. \tag{15}$$

Suppose there exists some saddle point $(u^*, v^*) \neq (x^*, y^*)$. Then $f(x^*, y^*) = f(u^*, v^*)$ must hold. Otherwise without loss of generality, assuming $f(x^*, y^*) < f(u^*, v^*)$, by the definition of saddle points, the fact $f(x^*, v^*) \leq f(x^*, y^*) < f(u^*, v^*)$ contradicts $f(u^*, v^*) \leq f(x^*, v^*)$.

Then, by (15) and Definition 2,

$$\begin{aligned} f(x^*, y^*) \leq f(\alpha x^* + (1 - \alpha)u^*, y^*) &< \alpha f(x^*, y^*) + (1 - \alpha)f(u^*, y^*) \\ &\leq \alpha f(x^*, y^*) + (1 - \alpha)f(u^*, v^*), \end{aligned}$$

which implies $f(x^*, y^*) < f(u^*, v^*)$, contradicting $f(x^*, y^*) = f(u^*, v^*)$. This completes the proof. $\square$

Finally, combining Lemmas 4 and 5 gives Lemma 2.

## D.2   Technical Lemmas

Before the convergence proof of Theorem 1, we need several technical lemmas.

**Lemma 6.** *(Relaxed triangle inequality) Let $v_1, \ldots, v_n$ be $n$ vectors in $\mathbb{R}^d$. Then,*

$$\left\| \sum_{i=1}^n v_i \right\|^2 \leq n \sum_{i=1}^n \|v_i\|^2.$$

**Lemma 7.** *Let $F_i(z) = (\nabla_x f_i(x, y), -\nabla_y f_i(x, y))$ where $z = (x, y)$. Under Assumption 1, $F_i(\cdot)$ is $\mu$-strongly monotone, $\forall i = 1, \ldots, m$, which means*

$$\langle F_i(z) - F_i(z'), z - z' \rangle \geq \mu\|z - z'\|^2, \quad \forall z, z' \in \mathbb{R}^{p+q}.$$

*Proof.* Let $g_i(x,y) = f_i(x,y) - \frac{\mu}{2}\|x\|^2 + \frac{\mu}{2}\|y\|^2$ and $G_i(z) = (\nabla_x g_i(x,y), -\nabla_y g_i(x,y))$, where $z = (x,y)$. From Assumption 1, it is obvious that $g_i(x,y)$ is convex-concave. Then, $F_i(z)$ is $\mu$-strongly monotone is equivalent to

$$\langle G_i(z) - G_i(z'), z - z' \rangle \geq 0, \ \forall z, z' \in \mathbb{R}^{p+q}.$$

Given the convex-concave property of $g_i(x,y)$, we have for any $z_1 = (x_1, y_1)$, $z_2 = (x_2, y_2)$,

$$
\begin{aligned}
g_i(x_2, y_1) &\geq g_i(x_1, y_1) + \langle \nabla_x g_i(x_1, y_1), x_2 - x_1 \rangle, \\
-g_i(x_1, y_2) &\geq -g_i(x_1, y_1) - \langle \nabla_y g_i(x_1, y_1), y_2 - y_1 \rangle, \\
g_i(x_1, y_2) &\geq g_i(x_2, y_2) + \langle \nabla_x g_i(x_2, y_2), x_1 - x_2 \rangle, \\
-g_i(x_2, y_1) &\geq -g_i(x_2, y_2) - \langle \nabla_y g_i(x_2, y_2), y_1 - y_2 \rangle.
\end{aligned}
$$

Adding these four inequalities gives

$$\langle \nabla_x g_i(x_1, y_1) - \nabla_x g_i(x_2, y_2), x_1 - x_2 \rangle - \langle g_i(x_1, y_1) - g_i(x_2, y_2), y_1 - y_2 \rangle \geq 0,$$

which essentially indicates $\langle G_i(z_1) - G_i(z_2), z_1 - z_2 \rangle \geq 0$. This completes the proof.

$\square$

**Lemma 8.** *For any $\mu$-strongly monotone and $L$-Lipschitz continuous operator $F(\cdot)$, there exists some $\lambda \in (0,1)$ such that given any $\eta \in (0, 2\mu/L^2)$,*

$$\|u - v - \eta(F(u) - F(v))\| \leq \lambda \|u - v\|, \ \forall u, v.$$

*Proof.*

$$
\begin{aligned}
\|u - v - \eta(F(u) - F(v))\|^2 &= \|u - v\|^2 + \eta^2\|F(u) - F(v)\|^2 - 2\eta\langle F(u) - F(v), u - v \rangle \\
&\leq \|u - v\|^2 + \eta^2 L^2 \|u - v\|^2 - 2\eta\mu\|u - v\|^2 \\
&= (1 - \eta(2\mu - \eta L^2))\|u - v\|^2
\end{aligned}
\tag{16}
$$

where Lemma 7 is used.

By setting $\lambda = 1 - \eta(2\mu - \eta L^2)$, we obtain $\lambda \in (0,1)$, $\forall \eta \in (0, 2\mu/L^2)$, which completes the proof. $\square$

### D.3 Proof of Theorem 1

In this section, we formally prove Theorem 1.

Define $z = (x,y)$, $F_i(z) = (\nabla_x f_i(x,y), -\nabla_y f_i(x,y))$, $F(z) = (\nabla_x f(x,y), -\nabla_y f(x,y))$. By definition, $F(z) = \frac{1}{m}\sum_{i=1}^{m} F_i(z)$. Denote $\text{Proj}_{\mathcal{Z}}(\cdot) = \text{Proj}_{\mathcal{X} \times \mathcal{Y}}(\cdot)$.

We focus on the updates within one outer iteration $t$ and may selectively drop the superscript $t$ in the following analysis for notational convenience. Then according to Algorithm 2, we obtain

$$
\begin{aligned}
z_{i,K} &= z_{i,0} - \eta \sum_{k=0}^{K-1} \left( F_i(z_{i,k}) - F_i(z^t) + F(z^t) \right) \\
&= z_{i,0} - \eta \sum_{k=0}^{K-1} F_i(z_{i,k}) + \eta K (F_i(z^t) - F(z^t)).
\end{aligned}
$$

Note that $z^t = \frac{1}{m}\sum_{i=1}^{m} z_{i,0}$ and $z^{t+1} = \text{Proj}_{\mathcal{Z}}\left(\frac{1}{m}\sum_{i=1}^{m} z_{i,K}\right)$, it yields

$$
\begin{aligned}
z^{t+1} &= \text{Proj}_{\mathcal{Z}}\left( \frac{1}{m}\sum_{i=1}^{m} z_{i,0} - \frac{\eta}{m}\sum_{i=1}^{m}\sum_{k=0}^{K-1} F_i(z_{i,k}) + \frac{\eta K}{m}\sum_{i=1}^{m}(F_i(z^t) - F(z^t)) \right) \\
&= \text{Proj}_{\mathcal{Z}}\left( z^t - \frac{\eta}{m}\sum_{i=1}^{m}\sum_{k=0}^{K-1} F_i(z_{i,k}) \right).
\end{aligned}
\tag{17}
$$

Then, we have

$$
\begin{aligned}
\|z^{t+1} - z^*\|^2 \quad \leq \quad & \left\| z^t - z^* - \frac{\eta}{m} \sum_{i=1}^{m} \sum_{k=0}^{K-1} F_i(z_{i,k}) \right\|^2 \\
= \quad & \|z^t - z^*\|^2 + \underbrace{\left\| \frac{\eta}{m} \sum_{i=1}^{m} \sum_{k=0}^{K-1} F_i(z_{i,k}) \right\|^2}_{\tau_1} \\
& \underbrace{-2\eta \langle \frac{1}{m} \sum_{i=1}^{m} \sum_{k=0}^{K-1} F_i(z_{i,k}), z^t - z^* \rangle}_{\tau_2}
\end{aligned}
\tag{18}
$$

where we use the fact that $\|\mathrm{Proj}_{\mathcal{Z}}(z_1) - \mathrm{Proj}_{\mathcal{Z}}(z_2)\| \leq \|z_1 - z_2\|$ and $z^* = (x^*, y^*)$.

Next, we will bound $\tau_1$. By noting $F(z^*) = 0$, we have

$$
\begin{aligned}
\tau_1 \quad = \quad & \frac{\eta^2}{m^2} \left\| \sum_{i=1}^{m} \sum_{k=0}^{K-1} F_i(z_{i,k}) \right\|^2 \\
= \quad & \frac{\eta^2}{m^2} \left\| \sum_{i=1}^{m} \sum_{k=0}^{K-1} (F_i(z_{i,k}) - F_i(z^*)) \right\|^2 \\
\overset{(a)}{\leq} \quad & \frac{\eta^2}{m} \sum_{i=1}^{m} \left\| \sum_{k=0}^{K-1} (F_i(z_{i,k}) - F_i(z^*)) \right\|^2 \\
\overset{(b)}{\leq} \quad & \frac{\eta^2 K}{m} \sum_{i=1}^{m} \sum_{k=0}^{K-1} \| F_i(z_{i,k}) - F_i(z^t) + F_i(z^t) - F_i(z^*) \|^2 \\
\overset{(c)}{\leq} \quad & \frac{\eta^2 K}{m} \sum_{i=1}^{m} \sum_{k=0}^{K-1} 2L^2 \left( \| z_{i,k} - z^t \|^2 + \| z^t - z^* \|^2 \right) \\
= \quad & \frac{2\eta^2 L^2 K}{m} \sum_{i=1}^{m} \sum_{k=0}^{K-1} \| z_{i,k} - z^t \|^2 + 2\eta^2 L^2 K^2 \| z^t - z^* \|^2
\end{aligned}
\tag{19}
$$

where (a) and (b) follow from the relaxed triangle inequality, and (c) follows from Assumption 2.

Then we will derive a bound for $\tau_2$.

$$
\begin{aligned}
\tau_2 &= -2\eta\langle\frac{1}{m}\sum_{i=1}^{m}\sum_{k=0}^{K-1}F_i(z_{i,k}), z^t - z^*\rangle \\
&= -2\eta\langle\frac{1}{m}\sum_{i=1}^{m}\sum_{k=0}^{K-1}F_i(z_{i,k}) - F_i(z^t) + F_i(z^t), z^t - z^*\rangle \\
&= -2\eta\langle\frac{1}{m}\sum_{i=1}^{m}\sum_{k=0}^{K-1}F_i(z_{i,k}) - F_i(z^t), z^t - z^*\rangle - 2\eta K\langle F(z^t), z^t - z^*\rangle \\
&\overset{(a)}{\leq} 2\eta\left\|\frac{1}{m}\sum_{i=1}^{m}\sum_{k=0}^{K-1}F_i(z_{i,k}) - F_i(z^t)\right\|\|z^t - z^*\| - 2\eta K\langle F(z^t), z^t - z^*\rangle \\
&\overset{(b)}{\leq} \frac{2\eta}{m}\sum_{i=1}^{m}\sum_{k=0}^{K-1}\|F_i(z_{i,k}) - F_i(z^t)\|\|z^t - z^*\| - 2\eta K\langle F(z^t), z^t - z^*\rangle \\
&\overset{(c)}{\leq} \frac{2\eta L}{m}\sum_{i=1}^{m}\sum_{k=0}^{K-1}\|z_{i,k} - z^t\|\|z^t - z^*\| - 2\eta K\langle F(z^t) - F(z^*), z^t - z^*\rangle \\
&\overset{(d)}{\leq} \frac{2\eta L}{m}\sum_{i=1}^{m}\sum_{k=0}^{K-1}\|z_{i,k} - z^t\|\|z^t - z^*\| - 2\eta\mu K\|z^t - z^*\|^2 \quad (20)
\end{aligned}
$$

where (a) follows from the Cauchy-Schwartz inequality; (b) follows from the triangle inequality; (c) follows from Assumption 2; (d) follows from Lemma 7.

From (19) and (20) we observe that both bounds are relevant to $\|z_{i,k} - z^t\|$, which indicates the drift between local models and the global model caused by multiple local updates before the communication. However, this drift can be bounded by the correction techniques of Algorithm 2:

$$
\begin{aligned}
\|z_{i,k+1} - z^t\| &= \|z_{i,k} - z^t - \eta(F_i(z_{i,k}) - F_i(z^t)) - \eta F(z^t)\| \\
&\leq \|z_{i,k} - z^t - \eta(F_i(z_{i,k}) - F_i(z^t))\| + \eta\|F(z^t)\| \\
&\leq \lambda\|z_{i,k} - z^t\| + \eta\|F(z^t)\|
\end{aligned}
$$

for some $0 < \lambda < 1$ with $0 < \eta < \frac{2\mu}{L^2}$ by Lemma 8. It further indicates for any $1 \leq k \leq K$,

$$
\begin{aligned}
\|z_{i,k} - z^t\| &\leq \lambda^k\|z_{i,0} - z^t\| + \eta k\|F(z^t)\| \\
&\leq \eta K\|F(z^t)\| \\
&\leq \eta K L\|z^t - z^*\| \quad (21)
\end{aligned}
$$

by noting $z_{i,0} = z^t$ and $\sum_{j=0}^{k-1}\lambda^j \leq k$.

Combining (18), (19) and (21) gives

$$
\begin{aligned}
\|z^{t+1} - z^*\|^2 &\leq (1 + 2\eta^2 L^2 K^2 - 2\eta\mu K)\|z^t - z^*\|^2 + 2(\eta L K)^4\|z^t - z^*\|^2 + 2(\eta L K)^2\|z^t - z^*\|^2 \\
&= \left(1 - 2(\eta\mu K - 2\eta^2 L^2 K^2 - \eta^4 L^4 K^4)\right)\|z^t - z^*\|^2.
\end{aligned}
$$

Let $h(\eta) = 2(\eta\mu K - 2\eta^2 L^2 K^2 - \eta^4 L^4 K^4)$. Given $0 < \eta \leq \frac{1}{2\mu K}$, $h(\eta) < 1$. Moreover,

$$
\frac{h(\eta)}{2\eta} = \mu K - 2\eta L^2 K^2 - \eta^3 L^4 K^4
$$

which is a monotonically decreasing function with respect to $\eta$ with $\lim_{\eta\to 0}\frac{h(\eta)}{2\eta} = \mu K > 0$ and $\lim_{\eta\to\infty}\frac{h(\eta)}{2\eta} = -\infty$. Then, we conclude that there exists some $\eta_1 > 0$ such that $h(\eta) > 0$, $\forall 0 < \eta < \eta_1$. By defining $\eta_0 = \min\{2\mu/L^2, 1/(2\mu K), \eta_1\}$, it yields $h(\eta) \in (0, 1)$, $\forall \eta \in (0, \eta_0)$. Defining $\rho(\eta) = 1 - h(\eta)$ completes the proof.

## D.4 Analysis of homogeneous local objectives

In this section, we analyze the convergence properties of FedGDA-GT under homogeneous setting. In fact, when all agents have identical objective functions, i.e., $f_i(x, y) = f(x, y), \forall i = 1, \ldots, m$, the convergence rate can be at least as $K$ times faster as that in Theorem 1, which is formally stated by the following proposition:

**Proposition 2.** *Suppose Assumptions 1, 2, 3 are satisfied and $f_i(x, y) = f(x, y), \forall i = 1, \ldots, m$. Let $\{(x^t, y^t)\}_{t=0}^{\infty}$ be a sequence generated by Algorithm 2. Then, choosing $\eta = \mu/L^2$, we have*

$$\|x^t - x^*\|^2 + \|y^t - y^*\|^2 \leq (1 - \kappa^{-2})^{Kt} \left(\|x^0 - x^*\|^2 + \|y^0 - y^*\|^2\right), \forall t = 0, 1, \ldots$$

*where $\kappa = L/\mu$ is the condition number of $f(x, y)$. Moreover, we have $1 - \kappa^2 \leq \rho(\eta), \forall \eta \in (0, \eta_0)$ where $\eta_0$ is defined in Theorem 1.*

*Proof.* As we stated before, Algorithm 2 reduces to conventional GDA under homogeneous setting. Then, for any $l \geq 0$, by the same techniques of Lemma 8, we have

$$\|z_{l+1} - z^*\|^2 \leq (1 - 2\eta\mu + \eta^2 L^2)\|z_l - z^*\|^2.$$

Setting $\eta = \mu/L^2$, $1 - 2\eta\mu + \eta^2 L^2$ reaches the smallest value, which is $1 - \kappa^{-2}$. Next, we prove that $\kappa^{-2} \geq h(\eta), \forall \eta > 0$. Note that

$$h(\eta) = 2(\eta\mu K - 2\eta^2 L^2 K^2 - \eta^4 L^4 K^4) \leq 2\eta(\mu K - \eta L^2 K^2) \leq \frac{1}{2}\kappa^{-2} < \kappa^{-2}.$$

Thus, we have $\rho(\eta) := 1 - h(\eta) \geq 1 - \kappa^{-2}, \forall \eta > 0$, which completes the proof. $\square$

To gain the intuition behind Proposition 2, we note that when $f_i(x, y) = f(x, y)$, $\nabla_x f_i(x, y) = \nabla_x f(x, y)$ and $\nabla_y f_i(x, y) = \nabla_y f(x, y)$. Then local updates in Algorithm 2 reduce to $x_{i,k+1}^{t+1} = x_{i,k}^{t+1} - \eta \nabla_x f(x_{i,k}^{t+1}, y_{i,k}^{t+1})$, similar for $y$. Since at the beginning all agents start at the same point $(x^0, y^0)$, it guarantees that for any $t$, $(x_{i,k}^t, y_{i,k}^t) = (x_{j,k}^t, y_{j,k}^t), \forall i, j \in \{1, \ldots, m\}$ and $\forall k = 0, \ldots, K - 1$. Thus, Algorithm 2 is equivalent to the centralized GDA in this homogeneous setting, where the global model is improved by $K$ times in one communication round.

# E  Analysis of generalization properties of minimax learning problems

In this section, we provide the formal proofs of the results in Section 4. Our proofs are based on the following technical tools.

**Definition 3.** *(Growth function) The growth function $\Pi_{\mathcal{H}} : \mathbb{N} \to \mathbb{N}$ for the hypothesis set $\mathcal{H}$ is defined by*

$$\Pi_{\mathcal{H}}(n) = \max_{\xi_1, \ldots, \xi_n} \left|\{(h(\xi_1), \ldots, h(\xi_n)) : h \in \mathcal{H}\}\right|$$

*where $\xi_1, \ldots, \xi_n$ are samples drawn according to some distribution.*

**Definition 4.** *(VC-dimension) The VC-dimension of hypothesis set $\mathcal{H}$ is defined by*

$$\text{VCdim}(\mathcal{H}) = \max\{n : \Pi_{\mathcal{H}}(n) = 2^n\}$$

*which measures the size of the largest set of points that can be shattered by $\mathcal{H}$.*

**Lemma 9.** *(Massart's lemma) Let $V \subseteq \mathbb{R}^n$ be a finite set such that $r = \max_{v \in V} \|v\|$. Then,*

$$\mathbb{E}_{\sigma}\left[\frac{1}{n} \sup_{v \in V} \sum_{j=1}^{n} \sigma_j v_j\right] \leq \frac{r\sqrt{2 \log |V|}}{n}$$

*where $v_j$ denotes the $j$th entry of $v$, each $\sigma_j$ is drawn independently from $\{-1, 1\}$ uniformly.*

**Lemma 10.** *(Sauer's lemma) Suppose the VC-dimension of hypothesis set $\mathcal{H}$ is $d$. Then for any integer $n \geq d$,*

$$\Pi_{\mathcal{H}}(n) \leq \left(\frac{en}{d}\right)^d.$$

We further introduce McDiarmid's inequality.

**Lemma 11.** *(McDiarmid's inequality) Let $X_1, \ldots, X_n$ are independent random variables with $X_i \in \mathcal{X}, \forall i = 1, \ldots, n$. Suppose there exist some function $f : \mathcal{X}^n \to \mathbb{R}$ and positive scalars $c_1, \ldots, c_n$ such that*

$$\left| f(x_1, \ldots, x_k, \ldots, x_n) - f(x_1, \ldots, x_k', \ldots, x_n) \right| \leq c_k$$

*for all $k = 1, \ldots, n$ and for any realizations $x_1, \ldots, x_n, x_k' \in \mathcal{X}$. Denote $f(X_1, \ldots, X_n)$ by $f(S)$. Then, for any $\epsilon > 0$,*

$$\mathbb{P}\left[f(S) - \mathbb{E}[f(S)] \geq \epsilon\right] \leq \exp\left(\frac{-2\epsilon^2}{\sum_{j=1}^n c_j^2}\right),$$

$$\mathbb{P}\left[f(S) - \mathbb{E}[f(S)] \leq -\epsilon\right] \leq \exp\left(\frac{-2\epsilon^2}{\sum_{j=1}^n c_j^2}\right).$$

Then, we are ready to give the proofs of results in Section 4.

### E.1   Proof of Theorem 2

Let $\mathcal{S} = \{\mathcal{S}_1, \ldots, \mathcal{S}_m\}$ be the collection of all local data sets. Given $y \in \mathcal{Y}$, define

$$\Phi(\mathcal{S}) = \sup_{x \in \mathcal{X}} \{R(x,y) - f(x,y)\}.$$

Let $\mathcal{S}' = \{\mathcal{S}_1', \ldots, \mathcal{S}_m'\}$ be another data collection differing from $\mathcal{S}$ only by point $\xi_{i,j}'$ in $\mathcal{S}_i'$ and $\xi_{i,j}$ in $\mathcal{S}_i$ for some specific $i$. Then,

$$
\begin{aligned}
\Phi(\mathcal{S}') - \Phi(\mathcal{S}) &= \sup_{x \in \mathcal{X}} \{R(x,y) - f'(x,y)\} - \sup_{x \in \mathcal{X}} \{R(x,y) - f(x,y)\} \\
&\leq \sup_{x \in \mathcal{X}} \{R(x,y) - f'(x,y) - (R(x,y) - f(x,y))\} \\
&= \sup_{x \in \mathcal{X}} \{f(x,y) - f'(x,y)\} \\
&= \sup_{x \in \mathcal{X}} \left\{ \frac{1}{mn} \sum_{j=1}^n l(x,y;\xi_{i,j}) - \frac{1}{mn} \sum_{j=1}^n l(x,y;\xi_{i,j}') \right\} \\
&\leq \frac{1}{mn} M_i(y).
\end{aligned}
$$

Applying McDiarmid's inequality gives that for any $c > 0$,

$$\mathbb{P}\left[\Phi(\mathcal{S}) - \mathbb{E}[\Phi(\mathcal{S})] \geq c\right] \leq \exp\left(\frac{-2c^2}{\sum_{i=1}^m \sum_{j=1}^n \left(\frac{M_i(y)}{mn}\right)^2}\right) = \exp\left(\frac{-2c^2 m^2 n}{\sum_{i=1}^m M_i^2(y)}\right).$$

Setting $\delta = \exp\left(\frac{-2c^2 m^2 n}{\sum_{i=1}^m M_i^2(y)}\right)$, we obtain $c = \sqrt{\sum_{i=1}^m \frac{M_i^2(y)}{2m^2 n} \log \frac{1}{\delta}}$. Then, with probability at least $1 - \delta$,

$$\sup_{x \in \mathcal{X}} \{R(x,y) - f(x,y)\} \leq \mathbb{E}\left[\sup_{x \in \mathcal{X}} \{R(x,y) - f(x,y)\}\right] + \sqrt{\sum_{i=1}^m \frac{M_i^2(y)}{2m^2 n} \log \frac{1}{\delta}}.$$

By similar techniques of [29], we have for any $y \in \mathcal{Y}$,

$$
\begin{aligned}
\mathbb{E}_{\xi \sim P}\left[\Phi(\mathcal{S})\right] &= \mathbb{E}_{\xi \sim P}\left[\sup_{x \in \mathcal{X}} \{R(x,y) - f(x,y)\}\right] \\
&= \mathbb{E}_{\xi \sim P}\left[\sup_{x \in \mathcal{X}} \{\mathbb{E}_{\xi' \sim P}\left[f'(x,y) - f(x,y)\right]\}\right] \\
&\leq \mathbb{E}_{\xi,\xi' \sim P}\left[\sup_{x \in \mathcal{X}}\{f'(x,y) - f(x,y)\}\right] \\
&= \mathbb{E}_{\sigma}\left[\mathbb{E}_{\xi,\xi' \sim P}\left[\sup_{x \in \mathcal{X}}\left\{\frac{1}{mn}\sum_{i=1}^{m}\sum_{j=1}^{n}\sigma_{i,j}(l(x,y;\xi'_{i,j}) - l(x,y;\xi_{i,j}))\right\}\right]\right] \\
&\leq 2\mathbb{E}_{\sigma}\left[\mathbb{E}_{\xi \sim P}\left[\sup_{x \in \mathcal{X}}\left\{\frac{1}{mn}\sum_{i=1}^{m}\sum_{j=1}^{n}\sigma_{i,j}l(x,y;\xi_{i,j})\right\}\right]\right] \\
&= 2\mathscr{R}(\mathcal{X},y)
\end{aligned}
$$

by noting $\xi$ and $\xi'$ are drawn from the same distribution and $\sigma$ is Rademacher variable. Thus, we have given $y \in \mathcal{Y}$, with probability at least $1 - \delta$,

$$
\sup_{x \in \mathcal{X}}\{R(x,y) - f(x,y)\} \leq 2\mathscr{R}(\mathcal{X},y) + \sqrt{\sum_{i=1}^{m}\frac{M_i^2(y)}{2m^2n}\log\frac{1}{\delta}}.
$$

Since $\mathcal{Y}$ is compact, every open cover of $\mathcal{Y}$ has a finite subcover. Then, we have $\left|\mathcal{Y}_\epsilon\right| < \infty$. Taking the union over $\mathcal{Y}_\epsilon$, it yields for any $x \in \mathcal{X}$ and $y \in \mathcal{Y}_\epsilon$, with probability at least $1 - \delta$,

$$
R(x,y) \leq f(x,y) + 2\mathscr{R}(\mathcal{X},y) + \sqrt{\sum_{i=1}^{m}\frac{M_i^2(y)}{2m^2n}\log\frac{|\mathcal{Y}_\epsilon|}{\delta}}.
$$

By the definition of $\mathcal{Y}_\epsilon$, for any $y \in \mathcal{Y}$, there exists a $y' \in \mathcal{Y}_\epsilon$ such that

$$
\begin{aligned}
R(x,y) - R(x,y') &\leq L_y\|y - y'\| \leq L_y\epsilon \\
f(x,y') - f(x,y) &\leq L_y\|y - y'\| \leq L_y\epsilon
\end{aligned}
$$

by Lipschitz continuity of $l$ in $y$. Thus, for any $\epsilon > 0$, $x \in \mathcal{X}$ and $y \in \mathcal{Y}$, with probability at least $1 - \delta$, the following inequality holds:

$$
R(x,y) \leq f(x,y) + 2\mathscr{R}(\mathcal{X},y) + \sqrt{\sum_{i=1}^{m}\frac{M_i^2(y)}{2m^2n}\log\frac{|\mathcal{Y}_\epsilon|}{\delta}} + 2L_y\epsilon,
$$

which completes the proof of Theorem 2.

### E.2  Proof of Corollary 1

From Theorem 2, it is obvious that with probability at least $1 - \delta$ for any $x \in \mathcal{X}$, taking the maximum of $\mathcal{Y}$ gives

$$
R(x,y) \leq \max_{y \in \mathcal{Y}}\left\{f(x,y) + 2\mathscr{R}(\mathcal{X},y) + \sqrt{\sum_{i=1}^{m}\frac{M_i^2(y)}{2m^2n}\log\frac{|\mathcal{Y}_\epsilon|}{\delta}}\right\} + 2L_y\epsilon.
$$

Since the above inequality holds for any $y \in \mathcal{Y}$, by again taking the maximum over $\mathcal{Y}$ on the left-hand side, we obtain for any $x \in \mathcal{X}$, with probability at least $1 - \delta$,

$$
\begin{aligned}
Q(x) &\leq \max_{y \in \mathcal{Y}}\left\{f(x,y) + 2\mathscr{R}(\mathcal{X},y) + \sqrt{\sum_{i=1}^{m}\frac{M_i^2(y)}{2m^2n}\log\frac{|\mathcal{Y}_\epsilon|}{\delta}}\right\} + 2L_y\epsilon \\
&\leq g(x) + 2\max_{y \in \mathcal{Y}}\{\mathscr{R}(\mathcal{X},y)\} + \sqrt{\max_{y \in \mathcal{Y}}\left\{\sum_{i=1}^{m}\frac{M_i^2(y)}{2m^2n}\right\}\log\frac{|\mathcal{Y}_\epsilon|}{\delta}} + 2L_y\epsilon
\end{aligned}
$$

which completes the proof.

### E.3 Proof of Lemma 3

First, for any fixed $y \in \mathcal{Y}$, define the growth function for the feasible set $\mathcal{X}$:

$$\Pi_{\mathcal{X}}^y(N) = \max_{\xi_1,\ldots,\xi_N} \left| \{(l(x,y;\xi_1),\ldots,l(x,y;\xi_N)) \ : \ x \in \mathcal{X}\} \right|,$$

where $N$ is the total number of samples drawn from the global distribution $P$. Essentially, the growth function $\Pi_{\mathcal{X}}^y(N)$ characterizes that given $y \in \mathcal{Y}$, the maximum number of distinct ways to label $N$ points.

Then for any $y \in \mathcal{Y}$, define a set $V^y$ of vectors in $\mathbb{R}^{mn}$ as

$$V^y = \{[l(x,y;\xi_{i,j})] : \xi_{i,j} \sim P, \ \forall i = 1,\ldots,m, \ j = 1,\ldots,n\}.$$

For any $v \in V^y$, we have

$$\|v\| = \sqrt{\sum_{i=1}^{m}\sum_{j=1}^{n} |l(x,y;\xi_{i,j})|^2} \leq \sqrt{\sum_{i=1}^{m} n M_i^2(y)}.$$

Then, by Massart's lemma, for any $y \in \mathcal{Y}$, it yields

$$\mathscr{R}(\mathcal{X},y) \leq \sqrt{\sum_{i=1}^{m} M_i^2(y)\frac{2\log|V^y|}{m^2 n}}.$$

Moreover, noting that for any $y \in \mathcal{Y}$, $|V^y| \leq \Pi_{\mathcal{X}}^y(mn)$ by the definition of $V^y$. Then, by applying Sauer's lemma, we have

$$\Pi_{\mathcal{X}}^y(mn) \leq \left(\frac{emn}{d}\right)^d$$

for all $mn \geq d$. By taking the maximum over $\mathcal{Y}$ on both sides of (22), we directly obtain (12).

## F  Code of the experiments

The datasets and the implementation of the experiments in Section 5 can be found through the following link: https://github.com/Starrskyy/FedGDA-GT.