# OpenReview forum: "A Communication-efficient Algorithm with Linear Convergence for Federated Minimax Learning"
_NeurIPS.cc/2022/Conference — NeurIPS 2022 Accept_

### Official Review · Reviewer_oPra · 2022-07-07

**Rating:** 7
**Confidence:** 4
**Soundness:** 4 excellent
**Presentation:** 3 good
**Contribution:** 3 good

**Summary:**

This paper studies the problem of min-max optimization in federated settings. The paper derives a Rademacher complexity bound for this setting, and then proceeds to analyze the local SGDA method in federated contexts. The paper shows that when using fixed learning rates and multiple local optimization steps, local SGDA does not converge to minimax optimal points (and give explicit examples of this in the quadratic setting in the appendix). The paper then proposes a method, FedGDA-GT, and shows that in certain settings, FedGDA-GT converges linearly to a minimax optimal point. Finally, the paper concludes with experiments comparing local SGDA and FedGDA-GT on synthetic data that show that FedGDA-GT can converge faster and to a point with lower loss.

**Questions:**

Some of these are distillations of questions from the section above.

*   Can Assumption 1 be relaxed at all?
*   Is there any benefit theoretically to setting $K > 1$ in Theorem 2?
*   Can you shed light on the form of $\rho(\eta)$ in Theorem 2?
*   In Proposition 1, can you concretize the trade-off between model accuracy and communication-efficiency you mention at the beginning of Section 4? For example, [1, 2] show that increasing $K$ for FedAvg does exactly this in quadratic settings.

Minor comments and questions:

*   The discussion before Figure 2(a) seems to suggest that the left plot is $\alpha = 0$, but the caption says $\alpha = 1$.
*   The bound quoted in (11) is incomplete. The dominant part of the convergence rate seems to be $O(c/\mu n T)$ for some $c$.
*   In Section 5.1, you seem to use the same learning rate for different methods and different values of $K$. Why is this valid?
*   In the empirical evaluations of Section 5.2, you use the same learning rate $\eta$ for different methods "for the sake of fair comparisons." Given that the methods are different, why is this fair?


#### References

[1] Malinovskiy et al, "From local SGD to local fixed-point methods for federated learning." ICML 2020.

[2] Charles & Konecny, " Convergence and accuracy trade-offs in federated learning and meta-learning." AISTATS 2021

**Limitations:**

There are no potential negative societal impacts that I can discern.

**Strengths And Weaknesses:**

I will preface this section with a note that I am not an expert in minimax optimization. Thus, I have taken the paper at face value in some of its discussion of related works on distributed minimax optimization.

That being said, I think this is a generally well-written paper that occupies an important point in a vein of works on federated optimization that the authors may not be aware of (as I will discuss below). The paper has 3 primary components (generalization bounds, convergence bounds, and empirical evaluation), and I have separated this discussion accordingly.

### Section 3 - Generalization bounds

The section gives a digestible introduction to Rademacher complexity, and gives corresponding generalization bounds for distributed minimax optimization that extend those of [Mohri et al., 2019]. The proofs seem sound and are interpretable.

That being said, this section is (in my opinion) one of the weaker portions of the paper. It feels disjointed from the rest of the work, especially as I found no observations that come out of the Rademacher complexity that influence any of the subsequent sections. For example, there is no facet of Corollary 1 that seems to influence the design of FedGDA-GT. Rather, the rest of the work is quite focused on the question of whether you converge to minimax optimal points of the training loss, so I'm not certain what the reader should take away from this section.

### Section 4 - Convergence bounds

In contrast to the above, this is the strongest part of the work. The observation that Local SGDA may not converge to minimax optimal points is a valuable one, and fits in nicely with a line of prior work in federated optimization (more on that below). Similarly, the FedGDA-GT method is an intuitively useful method that has nice convergence guarantees in certain settings. The proofs all seem sound, and the entire section gives a nice picture of federated minimax optimization.

The primary weakness of this section is something that is easily remedied, which will in turn make the paper stronger. This paper gives a minimax mirror to an important line of work in risk-minimization federated optimization that should be emphasized. Prior works, including [1, 2, 3] have all explored variations on a central theme: That Federated Averaging (FedAvg) does not converge to critical points of the empirical loss when using $K > 1$ local optimization steps. In that sense, Proposition 1 of this work is effectively a minimax version of this result, and I believe should be discussed as such.

Analogously, the solution to use gradient tracking kinds of techniques to remedy this issue is one that has appeared in multiple parts of the literature. Perhaps most fittingly, I would urge the authors to consider [4]. This work proposes a kind of gradient tracking version of FedAvg that has linear convergence in deterministic settings. In fact, the method FedGDA-GT is intuitively a minimax version of FedLin from [4]. The two methods share a very common framework, and have similar convergence results. Basically, I believe this work has done a good job of translating relevant work on objective inconsistency in federated learning to minimax settings, and I would like to see more of a discussion of this.

One other weakness of this section that is worth noting: It is unclear from Theorem 2 whether setting $K > 1$ in Algorithm 2 is useful.  In [4], the authors actually construct settings where $K = 1$ is optimal (by way of a lower bound). This is potentially a useful thing to add to this work if possible. Finally, I will note that while I would also love to see extensions to non-strongly-convex-strongly-concave settings, I do not think this is strictly necessary for acceptance.

### Section 5 - Empirical results

As mentioned above, these results are all on synthetic data. As such they succeed in illustrating the relevant properties of FedGDA-GT and Local SGDA that the authors want to illustrate. Moreover, I appreciated the authors' attempts to explore how heterogeneity affects the methods (and in particular, the observation that the methods are comparable in some settings).

While explorations on less synthetic data would improve the work, I do not think this is strictly necessary given the theoretical focus of the work.

### Review Summary

All in all, I think this is a good work that opens the way for many other analyses of federated minimax optimization. The strongest parts of this work are in Section 4. I would personally recommend the authors open this section up a bit more and include more about details about how it relates to federated risk-minimization works (eg. [1, 2, 3, 4]). In particular how this work presents an extension of these works to minimax settings. This related work also gives various avenues that could improve this work, including extensions to more general loss functions and possible lower bounds. While the generalization bounds and empirical evaluation are nice, they are not the primary draw of the work.

#### References

[1] Malinovskiy et al, "From local SGD to local fixed-point methods for federated learning." ICML 2020.

[2] Pathak & Wainwright, "FedSplit: An algorithmic framework for fast federated optimization." NeurIPS 2020.

[3] Charles & Konecny, " Convergence and accuracy trade-offs in federated learning and meta-learning." AISTATS 2021

[4] Mitra et al., "Linear convergence in federated learning: Tackling client heterogeneity and sparse gradients." NeurIPS 2021.

# Post author-feedback comments

I have read the other reviews of this work, and all the author comments. I still believe this paper should be accepted. While there are some discussions about the assumptions made regarding the minimax setting (eg. whether the strongly-convex-strongly-concave is too strong), I think that this misses the core contributions of the work:

1) A proof that Local SGDA with fixed learning rates will not converge to minimax-optimal points in federated settings when $K > 1$. Note that in particular, the linear convergence of Local SGDA only holds in non-federated settings: This work demonstrates that in federated settings it may not converge at all.

2) A new method that does converge linearly to minimax-optimal points in federated settings when $K > 1$ even with fixed learning rates.

As I mention in my review above, this mirrors a lot of theoretically interesting and provocative work on FL minimization over the last few years, but brings it to the realm of minimax optimization. I believe that this warrants publication (especially as the proofs are intuitive, easy to read, and correct). While the current organization of the work (with Rademacher complexity bounds coming first) may obscure these contributions slightly, this is eminently fixable.

Finally, I wish to note my strong belief that this is primarily a theoretical work, and should be reviewed accordingly. I think the experiments help validate the theory, and do not need to go beyond that. Similarly, I think that questions of the actual utility of this method in practical settings miss the point: This serves as a theoretical jumping off point for understanding why minimax optimization methods fail in FL settings, and how we can possibly remedy that. There is plenty of precedent in the literature for works of exactly that nature (including [1, 2, 3, 4] above).

I have increased my score to a 7 accordingly.

---

> ### Author Response · Authors · 2022-08-02
> **Response to Reviewer oPra's questions and minor comments**
>
> Response to Question 1: In our revised version, the twice-continuous differentiability assumption is removed (see Lemma 10 in Appendix E). In terms of strong-convexity-strong-concavity, it has been shown in (Zhang et al., 2019) to be a necessary condition for any first-order algorithm to achieve linear rate.
>
> Response to Question 2: A detailed discussion to this question is provided in the Response to "Comment 4".
>
> Response to Question 3: In Appendix E.3, we explicitly specify the form of $\rho(\eta)$, and conclude $\rho(\eta) \in (0,1)$ for suitable $\eta$. Specifically, from line 646 of Appendix E.3, we derive the expression of $\rho(\eta)$ and the corresponding conditions on $\eta$ to make $0 < \rho(\eta) < 1$. Please refer it for details.
>
> Response to Question 4: We would like to mention that for the federated minimax problem, the closed form of fixed points shown by Proposition 1 is generally hard to derive, even for the quadratic objective function. In Appendix D, we provide the closed form of fixed points with uncoupled quadratic functions for two agents and Figure 3 shows the trade-off between speed and accuracy.
>
> Response to Minor Comment 1: We double-check our result and it is exactly for $\alpha = 1$. Because of low heterogeneity, two algorithms perform similarly.
>
> Response to Minor Comment 2: We have fixed that issue in our revised version.
>
> Response to Minor Comments 3 and 4: Similar to most FL papers (e.g. [4]), we use the same learning rate for both algorithms, which essentially means under the same selection of stepsizes, FedGDA-GT outperforms Local SGDA.
>
> Reference:
>
> Junyu Zhang, Mingyi Hong, and Shuzhong Zhang. On lower iteration complexity bounds for the saddle point problems. arXiv preprint arXiv:1912.07481, 2019.

---

> > ### Comment · Reviewer_oPra · 2022-08-05
> > **A few notes on the minor comments**
> >
> > **Regarding minor comments 1 & 2:** Thank you for the feedback, and thank you for clarifying these.
> >
> > **Regarding minor comments 3 & 4:** I don't believe that most FL papers use the same learning rates for all algorithms (indeed, even in the FedLin paper you quote Figure 1 clearly shows multiple learning rates), or for all values of $K$. In fact, [1] shows that the learning rate needs to be tuned according to the number of local steps $K$, and that using different methods (eg. FedAvg versus FedProx) can make different learning rates optimal. While there are certainly papers that have fixed learning rates for different methods, I believe strongly that this is a bad practice both empirically and theoretically.
> >
> > This issue is not make-or-break for this work (which I view mainly as a theoretical treatise) but I do want to be clear that some discussion of this issue (eg. why does it still lead to fair comparisons in your setting) or at least positioning it as a weakness that could be remedied/studied better in future work would be a positive change to the paper.
> >
> > [1] Charles & Konecny, " Convergence and accuracy trade-offs in federated learning and meta-learning." AISTATS 2021.

---

> > > ### Author Response · Authors · 2022-08-05
> > > **Response to Reviewer oPra's notes on minor comments**
> > >
> > > Response to "Regarding minor comments 3 & 4": We really thank the reviewer for detailed feedback. We checked [1] in detail and do agree that learning rates should be tuned carefully according to $K$ to get optimal convergence rates for different algorithms. That is to say same stepsizes for different algorithms cannot lead to fair comparison. We will definitely avoid stating such thing and discuss its limitation in our final version. Since the effect of $K$ and stepsizes on the convergence rate needs more investigation and theoretical support (as that in [1]), we will take it as our future work and hope to provide guidance on finding optimal stepsizes afterwards.

---

> ### Author Response · Authors · 2022-08-02
> **Response to Reviewer oPra's comments**
>
> We thank the reviewer for the positive comments as well as constructive suggestions. Below, we discuss each of the reviewer's concerns, and explain how we plan to address them in the revised version of our manuscript.
>
> Comment 1: Section 3 feels disjointed from the rest of the work, since it does not influence any of the subsequent sections. Its takeaway is unclear.
>
> Response: We agree with the reviewer that the results in Section 3 do not influence subsequent sections. The main takeaway of Section 3 is to answer how many samples are needed to learn a model with good generalization performance for any distributed minimax learning problem? We recommend the reviewer to see the Response to "Comment 2" of Reviewer LKhi for more discussions.
>
> Comment 2: Proposition 1 of this work is effectively a minimax version of prior works [1][2][3], and should be discussed more.
>
> Response: We really appreciate the reviewer's valuable suggestions. We make detailed discussions as follows: firstly, we note that Proposition 1 is essentially aligned with [1][2][3]. In particular, if we set the variable $y$ be a constant, our minimax problem reduces to classical minimization problem and Proposition 1 characterizes the fixed-point behavior of FedAvg as those in [1][2][3]. Secondly, compared to [1][3] where fixed-point results are derived only under constant stepsizes, we would like to highlight that Proposition 1 also provides a direct interpretation why diminishing stepsizes are necessary for local updates algorithms without gradient tracking/error correction to converge exactly. Roughly speaking, as iteration index $t$ becomes large and stepsizes vanish towards zeros fast enough, the effect of local updates is negligible, which makes the fixed point not to deviate much from the optimal solution.
>
> Comment 3: FedGDA-GT is intuitively a minimax version of FedLin[4] and should be discussed more.
>
> Response: We appreciate the reviewer's recommendation on considering [4]. We have to highlight that although FedGDA-GT shares similar techniques as FedLin[4], the federated minimax optimization problem itself is much more complicated than conventional minimization problem. Even the property of optimal solutions of minimax problem remains an open problem until very recent work [59]. This also renders some challenges to the convergence analysis of FedGDA-GT.
>
> Comment 4: It is unclear from Theorem 2 whether setting $K>1$ in Algorithm 2 is useful, since [4] shows that $K=1$ is optimal with corresponding lower bound.
>
> Response: We do agree with the reviewer that setting $K=1$ might be worthy enough. From line 646 of Appendix E.3, the convergence rate of FedGDA-GT is influenced by $K$, where larger $K$ might cause a smaller optimal stepsize $\eta$. However, we have to mention that even for minimization problem, no FL algorithm has been able to establish tangible benefits of performing multiple local steps. For example, both Scaffold and FedLin employ stepsizes that scales inversely with local update steps $K$. In other words, it is a shared limitation among FL algorithms that local updates have significant benefits to the convergence rate. While this fact seems to undermine the value of FedGDA-GT, it is notable that FedGDA-GT actually reduces effort to find a "good enough" stepsize. Specifically, searching for learning rate is a practically hard work, due to complex structures of objectives. To ensure convergence, smaller stepsizes are preferred to be applied, which instead increases running time. In this sense, FedGDA-GT can converge much faster due to the effect of multiple local steps, compared to the case when $K=1$ with the same stepsize (see Figure 1). This advantage make stepsize selection much easier.
>
> In terms of the lower bound, we cite existing results to explain the convergence rate of FedGDA-GT is optimal. First, we note that when $K=1$, FedGDA-GT is the same as centralized GDA method. Second, the work (Zhang et al., 2019) has shown that the complexity lower bound of any first-order algorithm for strongly-convex-strongly-concave objectives is $\Omega(\log (1/\epsilon))$, which means the best rate is linear. Thus, FedGDA-GT is optimal.
>
> Reference:
>
> Junyu Zhang, Mingyi Hong, and Shuzhong Zhang. On lower iteration complexity bounds for the saddle point problems. arXiv preprint arXiv:1912.07481, 2019.

---

> > ### Comment · Reviewer_oPra · 2022-08-05
> > **A few brief points on the author feedback**
> >
> > Thanks for the detailed comments. I want to clarify a few of the points in my original review, especially with respect to the feedback you have given above.
> >
> > **Regarding Comment 1**: I have read Comment 2 that you posted to another reviewer above, and I still remain unconvinced that this section is giving the reader any particular insight into the rest of the paper. While I believe that the section is correct and novel, I ultimately still believe that the rest of the work is more interesting and should be emphasized any final presentation of this work. That being said, this is a minor concern, and is not really important in the grand scheme of things.
> >
> > **Regarding Comment 2**: I am glad you found those references useful. I wish to emphasize again though that many other works have already discussed why things like learning rate decay are necessary in federated learning settings. My point is simply (as I believe the authors agree with) that a more complete discussion of how this relates to previous work on minimization-oriented FL would benefit the paper from a readability perspective. I think that identifying trends of research and ideas and framing your new work in terms of these trends is a good thing, not something that diminishes the contributions of this work.
> >
> > **Regarding Comment 3**: My point in the original review was exactly what the authors stated: That FedLin and FedGDA-GT share similar gradient tracking techniques, at least at a high level. I understand the methods and use-cases are different, and I wish to emphasize that my comments were not trying to attack the novelty of your work or analysis. Rather, I think your work is novel, useful, and does a good job of showing the interested reader in how gradient tracking in standard FL can also be used for minimax FL settings. However, by adding a complete discussion of its parallels to FedLin, you make this paper much more approachable by a reader.
> >
> > For example, by pointing to the FedLin paper as an example of gradient tracking in standard minimization settings, readers with less background can look at that reference to first get a basic sense of how this works, and then come back to this work to understand how it can then be slotted into minimax optimization.
> >
> > **Regarding Comment 4**: The lower bound I am referring to is slightly different. As the authors mention, the work in (Zhang et al., 2019) is a lower bound for first-order optimization methods in centralized settings. It does not imply that we cannot do better in federated settings via the use of local updates. By contrast, the lower bound in the FedLin work is a lower bound specifically for FedLin (which is not a first-order optimization method in the standard sense) but helps illustrate that there are functions for which $K = 1$ is optimal. Hopefully this distinction makes sense.
> >
> > Regardless, I do not necessarily believe that this paper needs such a lower bound, but I believe it would be stronger with one. My original question was effectively asking, given the similarities between FedLin and FedGTA-GT, whether it is relatively simple to translate FedLin's lower bound to your setting.

---

> > > ### Author Response · Authors · 2022-08-05
> > > **Response to Reviewer oPra's more comments**
> > >
> > > We thank the reviewer for detailed reading our rebuttal and for further feedback. We will clarify the concerns raised by the reviewer in the following response.
> > >
> > > Response to "Regarding Comment 1": We do agree with the reviewer that Section 3 seems slightly disjoint with other parts of the paper. In our final version, we will try to shorten this section to put more emphasis on discussions of what the reviewer points out in "Regarding Comment 2" and "Regarding Comment 4". However, we still hope to clarify that the main purpose of Section 3 is to provide theoretical interpretation that what the best model we can achieve by only solving empirical minimax risk rather than the population one.
> > >
> > > Response to "Regarding Comment 2": We thank the reviewer very much for clarification and suggestions! As the reviewer stated, more discussions of related works on fixed-point behaviors of minimization FL problems could make the paper more comprehensive. We do agree with the reviewer and will definitely add a remark behind Proposition 1 in the camera-ready version to identify the relationship of our result with previous works (this is because of the page limit of current submission). In particular, we will state from the following aspects: (1) relevant analysis of minimization FL on fixed-point behaviors (e.g. [1][2][3]) would be discussed; (2) how Proposition 1 under minimax problems connects to previous results on minimization FL; (3) differences of our result compared to others on minimization problems would be clarified.
> > >
> > > Response to "Regarding Comment 3": We really appreciate the reviewer's further suggestions! In the camera-ready version, we will make a remark right behind our algorithm to clarify its connection with gradient-tracking-based FL algorithms, like Scaffold and FedLin. Specifically, we will first point out that how Scaffold and FedLin utilize gradient tracking techniques to resolve the incorrect fixed point issue in minimization-oriented FL, and then clarify how gradient tracking works in our algorithm by introducing more intuition.
> > >
> > > Response to "Regarding Comment 4": We thank the reviewer for clarification on the lower bound. We do think that a lower bound similar to that in FedLin can be derived when the objectives are uncoupled quadratic functions (i.e. $f_i (x,y) = x^T A_i x + b^T_i x - y^T C_i y - d^T_i y$). Then, we can directly use similar techniques of FedLin to get a similar lower bound in the sense that x and y updates of FedGDA-GT are independent, which thus can be treated equivalently as the update of minimization-oriented FL. We will include the formal result of lower bound of FedGDA-GT in the camera-ready version.

---

### Official Review · Reviewer_LKhi · 2022-07-11

**Rating:** 5
**Confidence:** 3
**Soundness:** 3 good
**Presentation:** 3 good
**Contribution:** 2 fair

**Summary:**

The paper characterizes the fixed-point properties of local SGDA with no gradient noise with multiple local steps and constant stepsize. Then FedGDA-GT, an improved Federated Gradient Descent Ascent method based on Gradient Tracking, is proposed. It is shown that FedGDA-GT converges linearly with O(\log\frac 1\epsilon) rounds of communication under smmoth strongly-convex-strongly-concave setting. Besides, the paper also analyzes the generalization of empirical minimax learning under distributed setting.

**Questions:**

1.	Is it possible to prove the convegence guarantee of FedGDA-GT with graident noise? Does linear convergence with constant stepsize still hold?
2.	What is the connection between section 3 and other sections despite the distributed minimax setting?


**Limitations:**

NA.

**Strengths And Weaknesses:**

Pros: This paper characterizes the fixed-point condition for local SGDA with multiple local steps with constant stepsize. The condition is not necessarily equivalent to the optimality condition, demonstrating local SGDA under this configuration is not necessarilty convergent to saddle point. The authors also provide an example to illustrate this fact.
	Under the same setting, FedGDA-GT can be shown to exhibit linear rate, which is faster than the sublinear rate of local SGDA. The gradient tracking is used to eliminate the client drift. The numerical experiment also provides evidence on the empirical benefit of FedGDA-GT.

Cons: The linear convergence rate of FedGDA-GT is derived under the setting without gradient noise. However, this is a distinct (and easier) scenario than the typical one in the federated learning framework where we only have access to the estimator of the gradient f_i. Whether the properties derived in this paper, i.e., linear rate with constant stepsize, still holds is unknown. Diminishing stepsize could be necessary to guarantee covergence and only sublinear rate is expected.
The paper also studies the generalization bounds on minimaxx learning problems in the distributed setting using Rademacher complexity. But its connection with other parts of the paper is unclear.

---

> ### Author Response · Authors · 2022-08-02
> **Response to Reviewer LKhi's comments and questions**
>
> We thank the reviewer for the positive comments as well as constructive suggestions. Below, we discuss each of the reviewer's concerns, and explain how we plan to address them in the revised version of our manuscript.
>
> Comment 1: The linear convergence rate of FedGDA-GT is derived under the setting without gradient noise. However, this is a distinct (and easier) scenario than the typical one in the federated learning framework where we only have access to the estimator of the gradient $f_i$. Whether the properties derived in this paper, i.e., linear rate with constant stepsize, still holds is unknown. Diminishing stepsize could be necessary to guarantee covergence and only sublinear rate is expected.
>
> Response: We agree with the reviewer that our method only considers deterministic objectives with exact gradient information. For general federated minimax optimization problem, local SGDA is, to the best of our knowledge, the only communication-efficient algorithm that allows multiple local updates with rigorous convergence guarantees. However, as Proposition 1 states, even under the ideal case with deterministic full gradients, local SGDA still cannot reach linear convergence to the exact solution, which is the main motivation of our algorithm. In this sense, FedGDA-GT is the first communication-efficient algorithm that guarantees correct fixed points with linear rate, which indicates a faster convergence rather than sublinear rate is expected. The case involving gradient noise and stochastic analysis of FedGDA-GT is very interesting and valuable, which will be our future work. Here, we could provide our insight on the convergence result when considering gradient noise. Intuitively, due to the gradient noise, we could expect that FedGDA-GT might finally converge linearly to some error neighborhood characterized by the variance of the gradient noise if constant stepsizes are applied. However, if exact convergence is required, diminishing stepsizes might be necessary in order to cancel the randomness caused by the gradient noise.
>
> Comment 2: The connection between Sections 3 and 4 is unclear.
>
> Response: We thank the reviewer for the concern. We argue that Section 3 essentially has a strong connection with Section 4 as we explain in the following. For the learning task, the original problem that we are interested in to find a model that can perform well on any possible data set, i.e., the ideal target is to find the optimal solution to (5). However, due to the unknown distribution of the data, the best we can do is to collect data samples and thus form the empirical minimax risk (1) that is solvable. Results in Section 3 indicates how well the model learnt from training set (i.e., the solution to (1)), is compared to the true model, that is, the solution to (5). In particular, Theorem 1 and Corollary 1 provide the sample complexity for learning a model empirically that is arbitrarily close to the true one. Based on these results, we could reasonably expect that problem (1) we solve (which is also the only thing solvable in practice) is a good enough approximation to the original problem (5) that we are interested in. After figuring our this problem, the next step is to design suitable algorithm that can solve (1) accurately and efficiently in a distributed way (and we particularly focus on the federated setting in this paper), which is the main topic of Section 4.
>
> To sum up, Section 3 first provides a positive answer that we can effectively learn a well-performed model by only accessing to data samples. Next, Section 4 gives an efficient algorithm to solve the problem.
>
> Response to Question 1: We thank the reviewer for the question. We believe the convergence guarantee on stochastic version of FedGDA-GT is possible and will include gradient noise and stochastic analysis in the future work. Here, we provide our insightful conjecture. By intuition, the gradient noise would bring some randomness to the trajectory of FedGDA-GT. Thus, we would expect that the expectation of the trajectory might either converge linearly to some error neighborhood characterized by noise variance or converge to the exact solution in a sublinear rate with diminishing stepsizes.
>
> Response to Question 2: We thank the reviewer for the question. We recommend the reviewer to see the Response to "Comment 2" for detailed interpretation.

---

> ### Author Response · Authors · 2022-08-09
> **A reminder of rebuttal review**
>
> Dear Reviewer LKhi,
>
> Thank you for reviewing our paper. We received a lot of constructive feedback from your valuable comments! According to your comments, we made detailed clarification and discussions in our response, which we hope you would find helpful. Since the deadline for author-reviewer discussion is quite close (Aug 9, 8PM UTC), we would really appreciate it if you could provide any further comments/suggestions for our response such that we could improve the paper accordingly.
>
> Yours sincerely,
>
> Authors of Paper 12588

---

### Official Review · Reviewer_gBj4 · 2022-07-12

**Rating:** 5
**Confidence:** 3
**Soundness:** 2 fair
**Presentation:** 1 poor
**Contribution:** 2 fair

**Summary:**

The paper is focused on minimax problems. First some generalization bounds are proposed via Rademacher complexity for the empirical minimax problem. Then, the paper focuses on federated learning for strongly-convex-strongly-concave minimax problems. A new algorithm  is proposed for which the convergence rate is provided.

**Questions:**

Qu1: How many communication rounds and what is the convergence rate of the proposed method compared with the results in [25] ? Using the same notations for clarity. Are the additional broadcasts from the server in FedGDA-GT accounted for ?

Qu2: What is the link between Section 3 presenting the Rademacher complexity and Section 4 presenting the proposed federated learning algorithm?


**Limitations:**

The limitatios of the propsoed method do not seem to be addressed in this paper.

Regarding negative societal impact, this work is fairly theoretical and there are no straightforward negative impact to be reported.

**Strengths And Weaknesses:**

Strengths

This paper investigates a federated learning approach for minimax problems. An interesting new algorithm is introduced, but which lacks a clear presentation.

Weaknesses

Overall, the paper lacks a clear focus and clear interpretations of the proposed algoritm and its main results in comparison with the state of the art.

1. The motivation behind the mathematical derivations in Section 3 are not clear. Some examples of its application could be provided.

2. The equation (11) does not seem to be accurate as 1/T^3 would mean a very fast convergence much better than 1/T. From [25], it seems that a rate convergence of O(1/(nT)) is achieved by local SGDA, but that the number of communications is either O(n) in the homogeneous setting or sqrt(nT) communication rounds in the heterogeneous setting.

3. Some further interpretation could be provided behind the result in Proposition 1. Especially, why this result implies that SGDA with a fixed point step-size cannot converge.

4. The discussion regarding local vs global minimax points right before section 4.2 is not clear especially since the objective is assumed to be strongly-convex-strongly-concave as per Assumption 1. This should be clarified.

5. The description of the novel proposed approach lacks in clarity. Some details behind the major differences between the SGDA and the proposed method FedGDA-GT should be provided after introducing the algorithm. It seems that some additional global information that is broadcasted by the server is used.

6. A more detailed discussion about the precise convergence rate and communication rounds behind the main result in Theorem 2 and in comparison to those of [25] should be provided before the Experimental section.

The main convergence result is not interpreted well enough and it is not clear why the proposed algorithm is superior than SGDA. Indeed, the convergence rate seems to be linear which is also the case for SGDA.

Regarding the communication rounds required by the proposed method, there is no clear discussion in the main body of the paper.

In the abstract the authors mention log(1/epsilon) communication rounds to reach an epsilon precision but this is nowhere to be found in the main body of the paper. In the conclusion it is mentioned log(1/epsilon) time, but this is very vague and unclear.

Comments post-rebuttal:

The authors have answered in a satisfactory manner to most raised comments. However, the paper still requires major changes to be accepted. Thus, the initial score is raised to 5.

---

> ### Author Response · Authors · 2022-08-02
> **Response to Reviewer gBj4's questions**
>
> Response to Question 1: We recommend the reviewer to see the Response to "Comment 6" for discussion of communication rounds. In terms of the server's broadcast, please see more details in the Response to "Comment 5".
>
> Response to Question 2: We thank the reviewer for the concern. We argue that Section 3 essentially has a strong connection with Section 4 as we explain in the following. For the learning task, the original problem that we are interested in to find a model that can perform well on any possible data set, i.e., the ideal target is to find the optimal solution to (5). However, due to the unknown distribution of the data, the best we can do is to collect data samples and thus form the empirical minimax risk (1) that is solvable. Results in Section 3 indicates how well the model learnt from training set (i.e., the solution to (1)), is compared to the true model, that is, the solution to (5). In particular, Theorem 1 and Corollary 1 provide the sample complexity for learning a model empirically that is arbitrarily close to the true one. Based on these results, we could reasonably expect that problem (1) we solve (which is also the only thing solvable in practice) is a good enough approximation to the original problem (5) that we are interested in. After figuring our this problem, the next step is to design suitable algorithm that can solve (1) accurately and efficiently in a distributed way (and we particularly focus on the federated setting in this paper), which is the main topic of Section 4.
>
> To sum up, Section 3 first provides a positive answer that we can effectively learn a well-performed model by only accessing to data samples. Next, Section 4 gives an efficient algorithm to solve the problem.

---

> ### Author Response · Authors · 2022-08-02
> **Response to Reviewer gBj4's comments (Comments 5 to 8)**
>
> Comment 5: The description of FedGDA-GT lacks in clarity. Major differences between the Local SGDA and the proposed method FedGDA-GT should be discussed. Additional global information broadcast by the server is used.
>
> Response: We appreciate the reviewer's feedback. We now explain how FedGDA-GT works as follows: in every iteration $t$, the server broadcasts its current model $(x^t, y^t)$ and computes global gradient $\nabla f(x^t, y^t)$, which is sent to all agents. Each agent updates in parallel local models using local gradients and correction terms $\nabla f(x^t, y^t)-\nabla f_i(x^t, y^t)$ for $K$ times. The server gathers local models for an averaging synchronization and projects it onto the feasible space. It is worth noting that the communication takes place only in the outer loop (indexed by $t$).
>
> Below our algorithm, we have stated the main difference between local SGDA and FedGDA-GT, that is, FedGDA-GT needs each agent to add a correction term $\nabla f(x^t, y^t) - \nabla f_i(x^t, y^t)$ by accessing the server's model. However, "there ain't no such thing as a free lunch". The gradient-tracking technique applied by FedGDA-GT causes additional global information exchange and an extra round of communication. In other words, FedGDA-GT resolves the incorrect fixed point issue of local SGDA by increasing communication (but not much) and using more information.
>
> Comment 6: A discussion about the convergence rate and communication rounds of Theorem 2 and in comparison to those of [25] should be provided.
>
> Response: We thank the reviewer for suggestions. We briefly explain the communication rounds of Theorem 2 and its comparison to [25]. Note that in order to obtain an $\epsilon$-approximation solution of (1), (i.e., we hope $\rho(\eta)^T \left( \Vert x^0 - x^* \Vert^2 + \Vert y^0 - y^* \Vert^2 \right) \le \epsilon$), communication rounds are $O(\log (1/ \epsilon))$. However, for local SGDA, Theorem 4.2 indicates that even for strongly-convex-strongly-concave cases, $O(1/\epsilon)$ communication rounds are needed, which is much bigger than $O(\log (1/ \epsilon))$. Thus, FedGDA-GT obtains exact convergence with lower communication.
>
> Comment 7: It is not clear why the proposed algorithm is superior than Local SGDA. Indeed, the convergence rate seems to be linear which is also the case for Local SGDA.
>
> Response: We argue that FedGDA-GT is superior than Local SGDA in the sense that our algorithm guarantees faster convergence and correct fixed points while Local SGDA does not. Theorem 2 states FedGDA-GT achieves linear convergence to correct fixed points, while Local SGDA's rate is $O(1/T)$, which is sublinear and much slower. This can also be verified by Figure 1 and results in Appendix D.
>
> Comment 8: There is no clear discussion in the main body of the paper that $\log(1/\epsilon)$ communication rounds are needed to reach an epsilon precision.
>
> Response: We clarify that essentially linear convergence reaches $O(\log (1/\epsilon))$ complexity. More discussions are provided in the Response to "Comment 6".

---

> ### Author Response · Authors · 2022-08-02
> **Response to Reviewer gBj4's comments (Comments 1 to 4)**
>
> We thank the reviewer for the positive comments as well as constructive suggestions. Below, we discuss each of the reviewer's concerns, and explain how we plan to address them in the revised version of our manuscript.
>
> Comment 1: The motivation behind the mathematical derivations in Section 3 are not clear. More examples should be provided.
>
> Response: We appreciate the feedback of the reviewer. The main motivation of Section 3 is to study the performance of the model trained on empirical objective (1) generalized on the population minimax risk (5). Since the population risk (5) is always unavailable, the best we can do is to train model with the data drawn from underlying distribution. Then Section 3 answers: how many data samples that we have to collect if the model trained empirically can perform well on any testing data set. Essentially, Theorem 1 and Corollary 1 state that if solution of (1) should be arbitrarily close to that of (5) (the true one), more data should be collected as shown in (8), (9). One application example is agnostic federated learning[13] and more are provided in Appendix A.
>
> Comment 2: The citation of equation (11) should be $O(1/T)$.
>
> Response: We have fixed it in our updated version.
>
> Comment 3: Some further interpretation could be provided behind the result in Proposition 1. Why Proposition 1 implies that Local SGDA with constant stepsize cannot converge.
>
> Response: We appreciate the reviewer's feedback. In particular, we focus on the case when $K=2$ and explain why Local SGDA converges to an incorrect fixed point. For $K=2$, the fixed point of local SGDA is the solution of $\frac{1}{m}\sum_{i=1}^m \nabla f_i(x^*, y^*) + \nabla f_i(x^* - \eta_x \nabla_x f_i(x^*, y^*), y^* + \eta_y \nabla_y f_i(x^*, y^*)) = 0$, which is not the same as $\frac{1}{m}\sum_{i=1}^m \nabla f_i(x^*, y^*) = 0$. We also recommend the reviewer to see Appendix D, where the exact form of the fixed point of local SGDA and numerical results are provided.
>
> Comment 4: The discussion regarding local vs global minimax points right before Section 4.2 is not clear especially since the objective is assumed to be strongly-convex-strongly-concave.
>
> Response: We clarify that the result of Proposition 1 and discussions before Section 4.2 are applicable for any objective functions. That is to say no matter what the objective is, local SGDA cannot guarantee exact convergence with constant stepsizes. For the strongly-convex-strongly-concave cases, we provided further discussions in Appendix D. Figure 3 shows our claim. And in discussions before Section 4.2, by local optimal point, we meant an optimal point with respect to $f_i$ instead of the global function $\sum_{i=1}^m f_i$.

---

> ### Author Response · Authors · 2022-08-09
> **A reminder of rebuttal review**
>
> Dear Reviewer gBj4,
>
> Thank you for reviewing our paper. We received a lot of constructive feedback from your valuable comments! According to your comments, we made detailed clarification and discussions in our response, which we hope you would find helpful. Since the deadline for author-reviewer discussion is quite close (Aug 9, 8PM UTC), we would really appreciate it if you could provide any further comments/suggestions for our response such that we could improve the paper accordingly.
>
> Yours sincerely,
>
> Authors of Paper 12588

---

### Official Review · Reviewer_urch · 2022-07-13

**Rating:** 4
**Confidence:** 4
**Soundness:** 2 fair
**Presentation:** 3 good
**Contribution:** 2 fair

**Summary:**

This paper considers a federated minimax optimization problem with compact constraints. The contents of this paper are threefold. First, under assumptions with respect to the maximization variable, the authors upper bound the stochastic minimax objective with the empirical minimax objective. They also provide an upper bound of Rademacher complexity of the stochastic minimax model. Second, a Federated (Fed) Gradient Descent Ascent (GDA) method based on Gradient Tracking (GT) (FedGDA-GT) is proposed. Under the assumption that the minimax problem is strongly convex and strongly concave, and other assumptions, the linear convergence of the generated sequence is proved. In the last part, experiments on quadratic minimax problems and robust linear regression problems are conducted.

**Questions:**

1.	In the experimental part, are the models satisfy the assumptions in section 4? Do they have the unique interior optimal solutions? If yes, does it mean the considered quadratic problems have closed forms?
2.	What’s the complexity that counts the inner iterations for the proposed method?
3.	What is the exact form of M(y) that is used as a key element in the upper bound of the Rademacher complexity? Can it be infinity when taking the maximum over y? If yes, does the upper bound in Corollary 2 and Lemma 2 become infinity?


**Ethics Review Area:**

["I don’t know"]

**Limitations:**

1.	The proposed algorithm needs to communicate with the server twice in each iteration, which can be inefficient.
2.	In applications of federated learning, the data size is huge and the proposed method requires computing the full gradient in each step. Thus, this method is expensive to apply. A stochastic variant of this method and its convergence analysis are not considered.
3.	The assumptions for the linear convergence of the proposed method are too strong to satisfy or check for the learning model. See more explanation in "strength and weakness" part 2.


**Strengths And Weaknesses:**

Strength:
1. The upper bound of Rademacher complexity for the general federated minimax model is new.
2. A gradient tracking skill is applied in the local updates of the proposed method. Thanks to this, in the deterministic case, the proposed method has sequential linear convergence under some assumptions.

Weakness:
1.	From my understanding, the upper bound of the Rademacher complexity can be obtained following similar assumptions and proofs in Section 4 [13] of this paper.
2.	In Section 4, the assumptions that the objective is strongly convex strongly concave, that the objective is twice continuously differentiable, and that the problem has at least an interior stationary point, are too strong. In centralized minimax problems, twice continuously differentiability is not usually assumed, but there are still methods that achieve linear convergence. Thus, the twice differentiability is not necessary and limits the scope of the applications of the convergence results. In addition, the assumption that there exists an interior stationary point is hard to check for many minimax models. Thanks to these ideal assumptions, the proofs for the linear convergence of the proposed method are classical and simple. In my opinion, these assumptions lack convincing application examples.
3.	The models in the experiments part are relatively simple, compared to those in [13]. No experiments are conducted on widely tested datasets such as Adult dataset, Fashion MNIST, and Language models. Since this paper claims to generalize the results in [13], I think experiments that apply the proposed method to the models in [13] should be conducted and compared.

---

> ### Author Response · Authors · 2022-08-02
> **Response to Reviewer urch's questions and limitations**
>
> Response to Question 1: Objectives in Section 5 satisfy assumptions. But, it is still hard to get the closed-form solution of quadratic problems, since both $x$ and $y$ variables are coupled.
>
> Response to Question 2: At each iteration $k$ of the inner loop, each agent only computes local gradients, which is efficient and can be done in parallel. Specifically, if we let $U_x$ and $U_y$ denote the time complexity of computing $\nabla_x f_i$ and $\nabla_y f_i$; $U_p$ denote the time complexity of projection. The total time complexity of the algorithm is  $\mathcal{O}((U_x + U_y)KT + U_p T)$.
>
> Response to Question 3: In the discussion before Section 4, we give one explicit form of $M_i(y)$ under agnostic federated learning[13], where $M_i(y) = m y_i M$ with positive constant $M$. The value of $M_i(y)$ cannot be infinite because of compactness of $Y$. In practice $M_i(y)$ is always continuous (e.g. $M_i(y) = m y_i M$ in [13]). Thus, $M_i(y)$ is bounded.
>
> Response to Limitation 1: We argue that our algorithm is indeed communication-efficient, because of multiple local updates with no communication at all. Since the inner iteration number $K$ is usually large, most updates of FedGDA-GT needs no communication and hence is communication-efficient. Moreover, the reason of twice communication at the outer loop is caused by gradient tracking technique, which is common in distributed optimization literature (Nedić et al., 2017). Moreover, FedGDA-GT is a significant improvement over conventional gradient tracking methods in the sense that gradient tracking algorithms require communication per iteration, while FedGDA-GT only involves communication after $K$ steps.
>
> Response to Limitation 2: The stochastic analysis of our algorithm is interesting and will be our future work. We also have to mention that, for federated minimax optimization problem, (to the best of our knowledge) no existing algorithms with multiple local updates can simultaneously guarantee linear rate and exact convergence even under deterministic cases, which is the major concern of this paper.
>
> Response to Limitation 3: We encourage the reviewer to see Responses to "Comment 2" and "Comment 3".
>
> References:
>
> Junyu Zhang, Mingyi Hong, and Shuzhong Zhang. On lower iteration complexity bounds for the saddle point problems. arXiv preprint arXiv:1912.07481, 2019.
>
> Angelia Nedić, Alex Olshevsky, and Wei Shi. Achieving geometric convergence for distributed optimization over time-varying graphs. SIAM Journal on Optimization, 27(4): 2597-2633, 2017.

---

> ### Author Response · Authors · 2022-08-02
> **Response to Reviewer urch's comments**
>
> We thank the reviewer for the positive comments as well as constructive suggestions. Below, we discuss each of the reviewer's concerns, and explain how we plan to address them in the revised version of our manuscript.
>
> Comment 1: The upper bound of the Rademacher complexity can be obtained following similar assumptions and proofs in [13] of this paper.
>
> Response: We appreciate the reviewer's concern. Despite the similar proof techniques, our results provide learning bounds that generalize [13], i.e., [13] studies particularly agnostic federated learning where feasible set $Y$ is a probability simplex, which is a special form of our problem. Please see our discussion at the end of Section 3. Moreover, our analysis places no restrictive assumptions on the problem settings (only compact and convex feasible sets are assumed) and hence fits for any distributed minimax learning problems.
>
> Comment 2: Twice continuous differentiability and strong-convexity-strong-concavity assumptions are too strong.
>
> Response: We remove twice continuous differentiability assumption in our revised version (in red). Please refer to Lemma 10 in Appendix E for detailed proofs (in red). In terms of the strong-convexity-strong-concavity assumption, we argue that this is a necessary condition to obtain linear convergence for any minimax problem with first-order algorithms (Zhang et al., 2019). In this sense, FedGDA-GT ahieves the best convergence rate.
>
> Comment 3: The assumption that there exists an interior stationary point is hard to check for many minimax models and lack application examples.
>
> Response: We agree with the reviewer that checking the existence of interior stationary points may not be easy. We also would like to remark that interior point methods is well adopted in conventional optimization. By adding a barrier function  for the constraints (e.g. -$p_1 \log(x)$ for a constraint of $x\geq 0$), the optimal solution would be repelled away from the boundary of the feasible sets. The coefficient $p_1$ in front of the barrier function is then diminished to approach 0 to recover the original optimal solution. For any positive $p_1$, the optimal solution is strictly interior. A similar approach could be applied to minimax problems studied here. In addition to introducing barrier functions, certain machine learning problems with regularization terms in objective and  norm constraints naturally enjoy the presence of interior point stationary points. This is because that the regularization term  pulls the optimal solution away from the boundary.
>
> Comment 4: More experiments should be conducted under real-world data sets in order to compare with [13].
>
> Response: We note that the main contribution of our paper is to propose the first algorithm with local updates that preserves optimal convergence rate and correct fixed points simultaneously, which is different from the focus of [13]. Due to the complex properties of real-world data, most experiments in [13] are implemented with non-convex-concave objectives, which do not lie in the scope of this paper and thus cannot be compared fairly. Moreover, no convergence rate of the algorithm in [13] is presented in their experimental results. To extend FedGDA-GT to real data sets, convergence analysis of non-convex-concave cases should be considered and we hope to solve it in the future.
>
> References:
>
> Junyu Zhang, Mingyi Hong, and Shuzhong Zhang. On lower iteration complexity bounds for the saddle point problems. arXiv preprint arXiv:1912.07481, 2019.
>
> Angelia Nedić, Alex Olshevsky, and Wei Shi. Achieving geometric convergence for distributed optimization over time-varying graphs. SIAM Journal on Optimization, 27(4): 2597-2633, 2017.

---

> ### Author Response · Authors · 2022-08-09
> **A reminder of rebuttal review**
>
> Dear Reviewer urch,
>
> Thank you for reviewing our paper. We received a lot of constructive feedback from your valuable comments! According to your comments, we made detailed clarification and discussions in our response, which we hope you would find helpful. Since the deadline for author-reviewer discussion is quite close (Aug 9, 8PM UTC), we would really appreciate it if you could provide any further comments/suggestions for our response such that we could improve the paper accordingly.
>
> Yours sincerely,
>
> Authors of Paper 12588

---

### Meta-Review · Area_Chair_CGwf · 2022-08-27

**Recommendation:** Accept
**Confidence:** Less certain

**Metareview:**

This paper has three main contributions: (1) a generalization bound for learning in adversarial learning frameworks such as GANs based on Rademacher complexity, (2) a proof that local SGDA with constant stepsize does not converge for these problems in federated settings and therefore does not achieve linear convergence, and (3) a new method which circumvents these issues.

Overall, the consensus was that result (1) was both somewhat underwhelming and also seemed somewhat disjointed from the paper. However, results (2) and (3) are compelling, and will be of interest to the federated learning community. This is especially true after the authors removed some of the technical assumptions that they required in the first version of the paper. The updated version of the paper could still use some cleaning up and/or reorganizing, however, I think that overall the paper is above the bar for acceptance.

**Award:**

No

---

### Decision · Program_Chairs · 2022-09-14

Accept